# Future Options of Molecular-Targeted Therapy in Small Cell Lung Cancer

**DOI:** 10.3390/cancers11050690

**Published:** 2019-05-17

**Authors:** Arik Bernard Schulze, Georg Evers, Andrea Kerkhoff, Michael Mohr, Christoph Schliemann, Wolfgang E. Berdel, Lars Henning Schmidt

**Affiliations:** Department of Medicine A, Hematology, Oncology and Pulmonary Medicine, University Hospital Muenster, 48149 Muenster, Germany; georg.evers@ukmuenster.de (G.E.); andrea.kerkhoff@ukmuenster.de (A.K.), mohrmic@ukmuenster.de (M.M.); christoph.schliemann@ukmuenster.de (C.S.); berdel@uni-muenster.de (W.E.B.)

**Keywords:** SCLC, anti-angiogenesis, apoptosis, epigenetics, targeted therapy

## Abstract

Lung cancer is the leading cause of cancer-related deaths worldwide. With a focus on histology, there are two major subtypes: Non-small cell lung cancer (NSCLC) (the more frequent subtype), and small cell lung cancer (SCLC) (the more aggressive one). Even though SCLC, in general, is a chemosensitive malignancy, relapses following induction therapy are frequent. The standard of care treatment of SCLC consists of platinum-based chemotherapy in combination with etoposide that is subsequently enhanced by PD-L1-inhibiting atezolizumab in the extensive-stage disease, as the addition of immune-checkpoint inhibition yielded improved overall survival. Although there are promising molecular pathways with potential therapeutic impacts, targeted therapies are still not an integral part of routine treatment. Against this background, we evaluated current literature for potential new molecular candidates such as surface markers (e.g., DLL3, TROP-2 or CD56), apoptotic factors (e.g., BCL-2, BET), genetic alterations (e.g., CREBBP, NOTCH or PTEN) or vascular markers (e.g., VEGF, FGFR1 or CD13). Apart from these factors, the application of so-called ‘poly-(ADP)-ribose polymerases’ (PARP) inhibitors can influence tumor repair mechanisms and thus offer new perspectives for future treatment. Another promising therapeutic concept is the inhibition of ‘enhancer of zeste homolog 2’ (EZH2) in the loss of function of tumor suppressors or amplification of (proto-) oncogenes. Considering the poor prognosis of SCLC patients, new molecular pathways require further investigation to augment our therapeutic armamentarium in the future.

## 1. Introduction

Worldwide, lung cancer is the main cause of cancer-related deaths [1] with increased mortality rates from an estimated one million cases in 2000 [2] to 1.59 million in 2012 [1]. Hence, the focus should be placed on early detection such as computed tomography screening [3] or prevention strategies including extensive smoking cessation programs [4]. Interestingly, the female proportion of lung cancer patients has increased, especially in Northern America [5], Northern Europe and Australia [1]. With a focus on histology, two major subtypes must be distinguished. With a frequency of up to 85%, non-small cell lung cancer (NSCLC) is the more frequent subtype [2,6]. In contrast, with a frequency of approximately 14%, small cell lung cancer (SCLC) is less frequent [7] but more aggressive. 

Initially, in 1879, SCLC was classified as lymphosarcoma deriving from the bronchial glands [8]. Following this, Barnard reported on the existence of large and small ‘oat cell’ carcinomas of the lung [9]. As a consequence, the term ‘small cell carcinoma’ of the lung was established. Recently, the 2015 World Health Organization (WHO) revision [6] categorizes neuroendocrine tumors such as SCLC, ‘large cell neuroendocrine carcinomas’ (LCNEC) and carcinoid tumors as genetic similar tumor entities [10,11]. Still, the ontogeny of small cell lung cancer initiating cells, sometimes termed ‘cancer stem cells’, is unclear. By whole-genome sequencing, George et al. identified tumor suppressor gene loci inactivation of ‘Tumor protein 53’ (Tp53) and ‘Retinoblastoma 1’ (RB1) as common mutations in SCLC [12]. While Meuwissen et al. associated protein depletion of tumor suppressors Tp53 and protein of Retinoblastoma 1 (pRB) with the malignant transformation of neuroendocrine cells [13], Sutherland et al. demonstrated that Tp53- and pRB-deficiency in genetically engineered mouse models (GEMMS) yielded malignant transformation of neuroendocrine lung cells and also alveolar type 2 cells [14]. Furthermore, typical neuroendocrine (i.e., smaller cells with, e.g., L-MYC+, ‘neural cell adhesion molecule’ [NCAM or CD56]+, Synaptophysin+) and atypical mesenchymal (i.e., larger cells with, e.g., Vimentin+, Nestin+, ‘homing cell adhesion molecule’ [HCAM or CD44]+) expression patterns were found after extraction of SCLC tissue from mice. Moreover, oncogenic Ras^V12^ expression is associated with a neuroendocrine to mesenchymal transition of the cell’s phenotype [15]. While L-MYC expression is more common in neuroendocrine small cells, C-MYC expression is associated with larger, neuroendocrine-low, NEUROD1+ SCLC cells [16]. Interestingly, George et al. detected NOTCH gene inactivation in 25% of human SCLC tissue. Vice versa, NOTCH pathway activation resulted in the suppression of neuroendocrine gene activity and reduced tumor growth [12]. In addition to these observations, strong ‘delta-like 3’ protein (DLL3) expression in immunostaining of SCLC tissue was found. DLL3 is a cell surface marker, which is internalized and inhibits the Notch proteins once it is activated [17,18]. 

Notably, about 3% to 10% of ‘epidermal growth factor receptor’ (EGFR)-mutant lung adenocarcinoma transform into SCLC [19]. Niederst et al. analyzed tissue samples of nine patients suffering from lung adenocarcinoma that transformed into SCLC following the development of EGFR inhibitor resistance. In each case, pRB loss was found [20]. Similar observations are described by Marcoux et al. in a cohort of n = 67 patients. Taken together, via an allele-specific polymerase chain reaction, next-generation sequencing, or whole-exome sequencing, they found relevant amounts of RB1 (79%), TP53 (58%) and ‘phosphatidylinositol 3-kinase catalytic subunit alpha’ (PIK3CA) (27%) mutations [19]. Sundaresan et al. postulated that the inactivation of the RB1 gene is associated with an increased overall mutational burden in SCLC tissues [21]. This genetic heterogeneity of SCLC is depicted in Table 1. Thereof, inactivating mutations of the ‘cAMP response element binding protein binding protein’ (CREBBP) gene locus are frequent: The latter gene transcriptional product, Crebbp, plays a crucial role as a ubiquitous transcriptional coactivator [12]. Furthermore, in many cases, the ‘phosphatase and tensin homolog’ (PTEN) locus [21], acting in the PI3K/AKT/mTOR pathway, is altered or dysfunctional. Other than that, the upregulated gene loci of oncogenic ‘avian-myelocytomatosis viral oncogene homolog’ (MYC) family and ‘fibroblast growth factor receptor 1’ (FGFR1) are frequently found [12]. 

In contrast to pulmonary adenocarcinoma, targetable activating kinase gene mutations are rare in SCLC. In due consideration of the results reported by Bordi et al. and Lu et al., EGFR mutations (exon 18–21 combined) were found in 6.6%, ‘B-Rapidly Accelerated Fibrosarcoma’ (B-RAF) mutations in 0.5% and ‘C-Mesenchymal-Epithelial Transition’ (C-MET) mutations in 4.4% [22,23]. Following the ‘Catalogue Of Somatic Mutations In Cancer’ (COSMIC) browser in SCLC, other than these, PIK3CA mutations were detected in 8% of the samples. However, over 90% of these were either nonsense or missense substitutions [24]. Regarding immunohistochemistry, receptor tyrosine kinase overexpression, other than c-KIT in LCNEC and SCLC was scarce [25]. Hence, inhibition of tyrosine kinases in SCLC might provide options for personalized medicine in the context of Next-Generation Sequencing (NGS) results for instance [26,27,28,29].

The rising incidence of female lung cancer patients [1,5] brings forth the question of whether sex steroids influence tumor progression and ultimately, patient’s outcome. Immunohistochemistry in NSCLC patients revealed both high estrogen receptor levels and aromatase enzyme levels in female patients. Here, aromatase enzyme upregulation was associated with reduced outcome [35]. However, there is no evidence for SCLC at present.

Inhibition of ‘programmed cell death 1’ (PD-1)/‘programmed cell death ligand 1’ (PD-L1) axis by nivolumab, pembrolizumab (both PD-1 inhibitors) or atezolizumab (PD-L1 inhibitor) augmented first- and second-line therapeutic options for NSCLC patients [36,37,38,39,40,41,42]. Recently, the PD-L1 inhibitor atezolizumab proved efficacy for first-line treatment of extensive-stage SCLC in combination with carboplatin and etoposide [43]. Similarly, the PD-1 inhibitor nivolumab yielded a therapeutic response in third-line SCLC therapy [44]. However, compared to second-line treatments, topotecan or amrubicin, nivolumab monotherapy application did not result in superior outcomes in SCLC [45]. Here, nivolumab plus the ‘cytotoxic T-lymphocyte-associated protein 4’ (CTLA-4) inhibitor ipilimumab is an advised therapy for refractory or relapsed SCLC [46]. Similar, pembrolizumab proved efficacy in refractory or relapsed SCLC [47] and is, therefore, recommended for SCLC treatment by the National Comprehensive Cancer Network (NCCN) [48].

In NSCLC, positive PD-L1 expression is associated with tumor response to immune-checkpoint-inhibition [36,49]. Likewise in SCLC, a positive PD-L1 expression (≥1%) is discussed to predict therapeutic response following immune oncologic treatment [46,47,50,51]. Additionally, in both SCLC and NSCLC, a high ‘tumor mutational burden’ (TMB) indicates durable treatment responses, especially following nivolumab and ipilimumab combination treatment [51,52,53]. Here, with regard to gender specifics, Wang et al. demonstrated that female NSCLC patients have a higher predictive value by TMB analysis in comparison to male patients [54]. However, these observations still need to be confirmed for SCLC.

Besides the PD-1/ PD-L1 blocking axis, another promising immune-stimulant SCLC treatment is maintenance therapy with the ‘toll-like receptor 9’ (TLR9) agonist, leftolimod [55]. Furthermore, possible immune therapeutic options might derive positive preclinical results by blocking the ‘integrin-associated protein’ CD47 on SCLC cells [56].

It is known that ‘prophylactic cranial irradiation’ (PCI) in SCLC is recommended in limited, as well as in extensive stage SCLC disease. Regarding its toxicity, experts have lately discussed its use in stage IV disease without ‘central nervous system’ (CNS) metastasis. In this specific situation, expert opinion limited the use to young and fit patients, that effectively responded to first-line therapy [57]. Other than that, thoracic irradiation in stage I–III disease is improving the prognosis, if implemented within 30 days of chemotherapy [58]. With regard to the ‘abscopal effect’, local irradiation and systemic chemotherapy in stage IV disease might promote the effect of concomitant immunotherapy with PD-1/PD-L1 inhibitors by inducing immunogenic cell death [59]. Hence, combination therapies of chemotherapy, irradiation and immunotherapy will be the focus of clinical research to further improve SCLC patients’ outcomes.

Other than treatment of the tumor entity by operation, irradiation, chemotherapy or immunotherapy, systemic risk factors influence the outcome of SCLC patients. As venous thrombosis and thromboembolism are risk factors for a dismal outcome, one study group focused on the additional use of low-molecular-weight heparin anticoagulants in SCLC treatment. Yet, the overall survival improved by the concomitant use of enoxaparin [60] nor was a biomarker established for predicting the risk of venous thrombosis and thromboembolism in SCLC patients [61]. Other than that, redox status and lipid metabolism were shown to be altered in lung cancer patients. Here, observational studies of lipid metabolism in lung cancer patients revealed reduced levels of high-density lipoprotein and apoprotein A1, as well as elevated levels of triglycerides [62]. Nevertheless, the anti-inflammatory and cholesterol-lowering effect of pravastatin plus cisplatin and etoposide did not result in a superior outcome in SCLC patients [63].

The underlying review focuses exclusively on potential molecular targets and pathways, which could offer therapeutic perspectives for SCLC treatment in the future.

## 2. Targeted Therapies in SCLC Treatment

While conventional chemotherapy is directed against all rapidly dividing cells (e.g., cells of the hematopoiesis or the gastrointestinal mucosa), targeted therapies focus on either the tumor cells or the peritumoral environment. Hereby, targeted therapies either interact with key tumor drivers (i.e., mostly specific mutations) that promote cell growth as well as immortality or address tumor-specific intra- or extracellular features.

Cancerogenesis is considered as a multistep process. Within this process, tumors acquire distinct biological capabilities also referred to as the hallmarks of cancer. Here, Hanahan and Weinberg defined six hallmarks of cancer: (1) To overcome apoptosis and cell death; (2) to perpetuate proliferation; (3) to evade growth inhibition; (4) to enable invasive and metastasizing growth; (5) to activate replicative immortality; (6) to generate tumor vasculature [64]. In contrast to the mere application of non-selective chemotherapies, the activation of the immune system or ‘targeted-based cancer therapy’ (TBCT) offers further therapeutic options. One major event in the development of targeted therapies was the introduction of monoclonal antibodies for cancer treatment [65]. Table 2 summarizes specific components of targeted therapy with a focus on tumor (micro-) environment, tumor cell biology and tumor genetics due to the application of ‘monoclonal antibodies’ (mAb) [66] and ‘small molecule inhibitors’ (SMI) [67,68,69]. 

### 2.1. Targeted Therapies in the Vascular System of SCLC

Both tumor growth and tumor spread depend on sufficient nutrition and oxygen supply, as well as on removal of metabolic waste. To guarantee these needs, neo-vascularization and neo-angiogenesis [64] can be induced by distinct molecular factors. Among these factors, ‘vascular endothelial growth factor’ (VEGF) and ‘basic fibroblast growth factor’ (bFGF) are key players in cancer progression. At present, we know of three options to attack the vascular system (i.e., anti-angiogenesis, vascular disruption and vascular infarction [97], see Figure 1).

#### 2.1.1. VEGF-Binding Bevacizumab, VEGF-R Antibodies and Multi-Tyrosine Kinase Inhibition in SCLC

Due to the intrinsic tyrosine kinase activity of its receptors, VEGF transmits cytoplasmic signaling and thus induces neo-vascularization [100]. Since the development of new tumor vessels is a crucial step in cancerogenesis, increased expression levels of VEGF are associated with poor prognosis both in SCLC [101,102] and in NSCLC [103,104,105]. To inhibit neo-vascularization, the anti-angiogenic soluble VEGF inhibitor, Bevacizumab, was constructed. This monoclonal antibody is already approved for the treatment of lung adenocarcinoma as first-line therapy [106,107,108]. However, in SCLC, the addition of Bevacizumab to standard chemotherapy did not improve overall survival rates [71], see Table 3. On the basis of impaired neo-vascularization, small-molecule ‘tyrosine kinase inhibitors’ (TKIs), which are directed against the VEGF receptor (VEGF-R) tyrosine kinase moiety (e.g., sunitinib, vandetanib or sorafenib), were developed. These drugs were first applied in NSCLC without major therapeutic benefit but increased toxicity profiles [98]. A placebo, controlled phase II study of sunitinib maintenance therapy (i.e., PDGFRα- and β-, VEGFR1-,2-,3-, RET-, c-KIT- and FLT3-inhibitor) for extensive-stage SCLC without progression after four to six cycles of platin and etoposide resulted in a significantly longer progression-free survival (PFS) (3.7 months vs. 2.0 months). However, overall survival (OS) was not improved [73]. A phase II sunitinib monotherapy both for chemosensitive relapsed and for chemo-naïve SCLC patients closed due to high hematotoxicity and clinical lack of solitary efficacy after enrolment of nine patients [75], see Table 3. Other than sunitinib, in a randomized, placebo-controlled first-line phase II trial, the impact of the multi-tyrosine kinase inhibitor, vandetanib, (i.e., VEGFR3-, EGFR- and RET-inhibition) in combination with cisplatin and etoposide was investigated. Here, OS for vandetanib vs. placebo (i.e., 13.24 months vs. 9.23 months, *p* = 0.458) and objective response rate (ORR) (50% vs. 65%) did not differ significantly [74], see Table 3. Although VEGF-R2 inhibition with Ramucirumab is an accepted therapy for relapsed NSCLC [109], there is little evidence in SCLC. At present, there are ongoing studies which evaluate both the VEGF-R2 TKI apatinib (NCT02995187) and the multi-tyrosine kinase inhibitor, chiauranib, for refractory SCLC patients (NCT03216343). In contrast, in a genetically engineered, murine SCLC model with Retinoblastoma 1 (pRB)- and Tp53-protein deficiency, programmed cell death-ligand 1 (PD-L1) and VEGF signaling disruption yielded an improved tumor response and prolonged survival. Without doubt, VEGF-signaling also has costimulatory activity regarding exhausted T-cells via T-cell immunoglobulin and the mucin 3 domain (TIM3). Potentially, VEGF signaling inhibition could overcome PD-1/PD-L1 axis resistance in SCLC patients [110].

#### 2.1.2. Anti-Angiogenesis in FGFR1-Amplified SCLC

Besides the VEGF-signaling pathway, ‘fibroblast growth factor receptor’ (FGFR) activation by bFGF is another cornerstone in neo-angiogenesis [111]. The FGFR1 gene is frequently overexpressed in about 5.6% of SCLC patients [32]. In approximately 22% of the patients, FGFR1 overexpression is more frequent in NSCLC with squamous cell histology [112]. However, in both cases, it harbors a potential therapeutic impact. High FGFR expression on the SCLC cell surface is inversely correlated with chemosensitivity [112]. Whereas overexpression of bFGF is associated with reduced prognosis in SCLC, in advanced NSCLC no adverse effects were found [113]. The FGFR is a cell-surface receptor tyrosine kinase (RTK), and has, amongst others, two clinically relevant downstream signaling pathways: The ‘rat sarcoma’ (RAS)/‘rapidly accelerated fibrosarcoma’ (RAF)/‘mitogen-activated protein kinase’ (MEK/MAPK) signaling, as well as the ‘phosphatidyl-inositol 3 kinase’ (PI3K)/protein kinase B (PKB/AKT)/ ‘mammalian target of rapamycin’ (mTOR) signaling [114]. 

A phase II study of the multi-tyrosine kinase inhibitor, nintedanib, (including VEGFR-, FGFR and PDGFR-inhibition) as monotherapy in relapsed SCLC did not improve prognosis, see Table 3. The PFS was one month and the objective response rate was only 5% [72]. Other multi-tyrosine kinase inhibitors such as lucitanib [115] and the pan-FGFR-inhibitor erdafitinib (AZD4547) were investigated in phase I studies [116] in solid tumors without any advantage in SCLC. A promising agent in irreversible pan-FGFR-inhibition is TAS-120 [117]. For this drug, a phase I study in solid tumors including SCLC is ongoing. Potentially, there is a beneficial treatment option for FGFR1-amplified SCLC patients (NCT02052778). In a murine model for hepatocellular carcinoma, response following the application of ^125^iodine-labeled bFGF monoclonal antibody (mAb) was reported [118]. In vitro, another study even demonstrated the therapeutic impact of a bFGF mAb on H223 SCLC cell cultures both with and without irinotecan [119]. Furthermore, the bFGF-directed monoclonal antibody, E12, decreased tumor growth and migration by E-cadherin upregulation in murine Lewis lung carcinoma [120].

#### 2.1.3. Anti-Angiogenesis in SCLC by Imides 

Another interesting treatment option due to the emerging insensitivity towards bFGF [121] and VEGF could be induced by imides [122]. This therapeutic principle is already applied for the therapy of multiple myeloma [123], myelodysplastic syndrome and specific types of lymphomas [124]. Nevertheless, the placebo-controlled phase III study discovered no significant survival benefit upon the additional application of thalidomide following response to two cycles of platinum-based chemotherapy, see Table 3. Here, the insignificant but impressive prolongation of the overall survival of three months in the thalidomide cohort (i.e., 11.7 months vs. 8.7 months, *p* = 0.160) requires careful consideration in light of the study design with its necessary therapeutic response after two cycles of non-standard care therapy [76]. Improved survival rates were found only for the subgroup of patients with an ECOG performance status of 1 or 2 [76]. Similarly, another phase III placebo-controlled trial evaluated the benefit of thalidomide in combination with carboplatin and etoposide for SCLC patients with both limited and extensive disease stages. Whereas OS did not differ significantly in the full study collective (*p* = 0.28), treatment-related toxicity (i.e., neuropathy, rash and constipation) were more common in the thalidomide treatment arm [77], see Table 3. Likewise, pomalidomide, in combination with cisplatin and etoposide, did not yield an improved prognosis in extensive-disease-stage SCLC. However, each patient presented grade 3 or higher toxicity and subsequent consequences included pulmonary embolism, cerebral ischemia and sepsis leading to death [78], see Table 3. As a result, imides are no longer under investigation for SCLC treatment.

#### 2.1.4. Vascular Disrupting Agents in SCLC

Apart from therapies against pro-angiogenic factors, vascular disrupting agents (VDAs) could increase the therapeutic impact of cytotoxic chemotherapy and irradiation by attacking the existing vasculature [125]. Among VDAs, two main classes are distinguished: Small molecules that destabilize microtubules and induce cytokines, and ligand-directed drugs that attack distinct target antigens in tumor blood vessels. The most prominent representative of ligand-directed VDA is bavituximab, which recognizes a specific membrane phospholipid in the tumor endothelium. Once the membrane phospholipid is bound to bavituximab, vessel destruction is initiated by antibody-dependent cellular cytotoxicity [126]. Recently, a placebo-controlled phase III trial investigated its role in combination with docetaxel for patients with advanced NSCLC. However, in clinical practice, no additional benefit of bavituximab (OS 10.5 months) in comparison to placebo (OS 10.9 months, *p* = 0.533) was discovered [127]. To our knowledge, there is no published data for the successful application of any other VDA in SCLC so far.

#### 2.1.5. Vascular Infarction in SCLC

Since angiogenesis can be induced by the membranous glycoprotein aminopeptidase N (APN, CD13) [128], tumor growth and metastasis are influenced [129] in various cancer types [130,131,132,133], including NSCLC [134,135]. Therapeutically, vascular infarction can be induced by the application of ‘truncated tissue factor’ (tTF) coupled to ‘asparagine–glycine–arginine’ (NGR) for instance. The truncated tissue factor induces the extrinsic coagulation process by activating factor X [99]. As demonstrated before, CD13 serves as a target structure for NGR-peptides and makes it a suitable candidate for targeted therapy [128]. Recently, preclinical experiments demonstrated antitumor effects for this therapeutic principle in a variety of human xenograft models with different histology, such as colon, breast, melanoma, soft tissue sarcoma [136,137,138,139,140,141,142], NSCLC [135] and SCLC [143]. At present, tTF-NGR is under investigation in a phase I trial (NCT02902237).

### 2.2. Targeting Transcription and Epigenetic Factors in SCLC

#### 2.2.1. Curaxins in Tp53 and Notch1-Deficient SCLC

Gurova et al. demonstrated that quinacrine, an antimalarial drug, and 9-aminoacridine (9AA), were able to induce Tp53 and deplete NFκB in renal cell carcinomas. In a mouse model, both substances decreased tumor growth with an efficacy similar to that of 5-fluorouracil [144]. Against this background, so-called ‘curaxin’ agents were developed. The application of these intercalating drugs leads to structural DNA changes. The histone chaperone transcription factor ‘FAcilitates Chromatin Transcription’ (FACT) binds to curaxin-altered DNA sequences and is trapped there. This reduces signaling pathways such as NFκB for instance [145]. Besides this mechanism, casein kinase 2 (CK2) can bind to the FACT-DNA-curaxin complex and thus mediate phosphorylation of Tp53 at Ser392 site. As a result, Tp53 activity is enhanced due to its reduced degradation of Tp53 via mouse double minute 2 homolog (Mdm2) [145] (see Figure 2). 

Recently, De et al. described synergistic effects in the treatment of SCLC patient-derived xenograft mouse models with the curaxin CBL0137 plus Cisplatin [146]. As NOTCH activation and downstream signaling suppress tumor growth in SCLC [12], in vitro studies of De et al. with H82 and H526 cells also revealed eventual upregulated NOTCH1 mRNA, especially in CD133+ tumor stem-like cells after treatment with CBL0137, by inhibiting the SP3-transcription repressor from binding to the NOTCH1 promotor region [146]. However, the effect of curaxins on Tp53 still requires further investigation in Tp53-inactivated SCLC. Currently, a phase I trial of CBL0137 in advanced solid neoplasms and glioblastoma (NCT01905228) is ongoing.

#### 2.2.2. Other Transcription Inhibitors in SCLC

Lurbinectedin, a selective inhibitor of trans-activated RNA polymerase II transcription, was tested in SCLC. A phase I trial in refractory and relapsed SCLC patients treated with doxorubicin plus lurbinectedin (PM01183) brought forth promising results especially, in chemosensitive relapse (ORR 91.7%, PFS 5.8 months) [147]. Additionally, in second-line treatment, doxorubicin plus lurbinectedin was confirmed to achieve response rates comparable to first-line cisplatin and etoposide [148]. Due to a reasonable side-effect profile, selective inhibitors of trans-activated RNA polymerase II might harbor potential in SCLC first-line treatment.

#### 2.2.3. Notch2/3 Inhibition in SCLC

Besides Notch1 tumor suppressor function in SCLC, Notch2 and Notch3 have oncogenic potential in SCLC [149]. Considering this, the Notch2/3 inhibitor, tarextumab (OMP-59R5), was investigated in a phase I study together with various solid tumors [150]. Phase Ib data on untreated, extensive-stage SCLC in combination with cis- or carboplatin and etoposide demonstrated an ORR of 73% with a median overall survival of 16 months for patients, receiving ≥12.5 mg/kg tarextumab every 21 days [151]. However, a randomized, placebo-controlled, phase II trial of tarextumab in combination with cis- or carboplatin and etoposide failed to prove efficacy in untreated SCLC patients with extensive disease stages. PFS was 5.5 months, both for tarextumab and placebo-treated patients [79], see Table 3.

#### 2.2.4. HDAC Inhibition in Crebbp-Deficient SCLC

The ‘cAMP response element binding protein’ binding protein (Crebbp) is an acetyltransferase with influence on the chromatin accessibility. Especially, the acetylation of the associated histone H3K27 has an influence on promoter activity and transcriptional processes [152]. Considering the high frequency of CREBBP inactivation in SCLC, the function of Crebbp in neuroendocrine tumors was evaluated by Jia et al. Both in vitro and in vivo CREBBP inactivation accelerated SCLC growth [153]. In contrast, Crebbp loss favored the epithelial to mesenchymal transition (EMT) and E-cadherin reduction. Hence, Crebbp is suggested to act as a tumor suppressor in SCLC [153]. Interestingly, the use of histone deacetylase inhibitor (HDACi) pracinostat, yielded increased levels of acetylated H3K27 (H3K27Ac) and E-cadherin, eventually overcoming the effects of a lack of Crebbp [153]. A placebo-controlled phase III study with pracinostat in combination with azacitidine in acute myeloid leukemia (NCT03151408) was performed; however, there is no clinical evidence for its effectiveness in SCLC.

#### 2.2.5. Aurora Kinase Inhibition in pRB-Deficient and MYC-Amplified SCLC

Aurora kinases regulate relevant steps in the cell cycle by phosphorylating key players. Here, ‘Aurora A kinase’ (AURKA) regulates early mitotic prophase and prometaphase by phosphorylation of microtubule-associated proteins, the emergence of the bipolar spindle apparatus, as well as the G_2_-M arrest. Moreover, ‘Aurora B kinase’ (AURKB) as a part of the chromosomal passenger complex influences chromosome alignment in the metaphase and during the cytokinesis [154]. Helfrich et al. investigated the AURKB inhibitor, barasertib, in c-MYC, MYCL1 and MYCN-amplified SCLC cell lines. Here, cell lines with strong C-MYC expression displayed a 16-fold higher sensitivity to barasertib compared to the other cell lines. However, vulnerability to barasertib was neither associated with MYCL1- nor with MYCN amplification [155].

Independent from c-MYC expression, Oser et al. investigated the use of AZD2811, an AURKB inhibitor, in RB1-deficient SCLC cell lines and xenograft mouse models [156]. The expression of pRB in healthy cells yields cell cycle control by disrupting E2F-complex-mediated transcription [157]. Hence, loss of pRB itself strengthens the upregulation of gene sets with influence on the mitotic spindle apparatus and chromosomal segregation process. In pRB proficient cells, AURKB inhibition did not relevantly affect cell growth and apoptosis, whereas in pRB-deficient tumors AURKB inhibition exacerbated polyploidy and ultimately, resulted in high apoptotic rates [156]. 

Since RB1 inactivation accounts for about two-thirds of SCLC patients [21] and amplification of a MYC family gene was found in about one-fifth [31] of SCLC cases, two studies reported on these genetic changes as prone conditions for the use of aurora kinase inhibitors [16,156].

Here, Melichar et al. evaluated alisertib monotherapy in a phase II study by performing a comparison of n = 48 relapsed or refractory SCLC patients. The objective response rate (ORR) on monotherapy was impressively high with 21%, yet another 33% reached a stable disease state [80], see Table 3. Owonikoko et al. performed a second-line, placebo-controlled phase II study of alisertib in combination with paclitaxel, revealing a significantly prolonged PFS of 101 days compared to 66 days for paclitaxel and placebo. Alas, OS was not improved [81], see Table 3, and both studies did not focus on patients’ c-MYC or RB1 mutational status.

#### 2.2.6. Hedgehog-Cascade Inhibition in SCLC

Disease progression in genetically engineered SCLC mouse models with Tp53 and Rb1 mutations revealed upregulated Hedgehog signaling. Via ligand binding of ‘Sonic hedgehog’ (Shh) to ‘Patched’ (PTCH1), a downstream inhibition of ‘Smoothened’ (SMO) is induced. On the contrary, deletion of SMO halted SCLC growth [158]. The latter itself activates Gli-family zinc-finger transcription factors. Here, SMO and Gli-inhibitors can interrupt the canonical Hedgehog signal cascade and interfere with tumor growth [159]. Interestingly, Park et al. found that monotherapy with the SMO-inhibitor sonidegib (LDE225) itself did not suppress tumor growth but improved the efficacy of platinum-based chemotherapy [158]. 

A phase I trial of n = 14 patients, treated with sonidegib plus cisplatin and etoposide in extensive-disease stage SCLC revealed high overall response rates of 79% (all partial remission), while the disease control rate was 100% [82]. However, a phase II randomized, three armed study of SCLC patients with extensive disease, who were treated with Cisplatin and Etoposide alone, in combination with SMO-inhibiting vismodegib or with the ‘insulin-like growth factor 1 receptor’ (IGF-1R) antagonist cixutumumab did result in improved PFS or OS for neither vismodegib nor cixutumumab (i.e., OS 8.8 months vs. 9.8 months vs. 10.1 months) [83], see Table 3. Due to these data, there is no further clinical investigation of SMO-inhibitors in SCLC. Still, few data exist on Gli1-downregulation and cancer stem-like cell killing via arsenic trioxide, possibly revealing another hedgehog-targeted anti-tumor therapy [160].

#### 2.2.7. SOX2-Directed Therapy in SCLC

The oncogene ‘sex determining region Y (Sry)-related high-mobility (HMG) box 2’ (SOX2) is frequently upregulated in SCLC [33] and is associated with poor prognosis [161]. Along with CD133 and Oct4, it is seen as a cancer stem-like cell marker in SCLC [160] and interacts with MYC, Notch and Wnt proteins [162]. It is regulated by the canonical hedgehog signal cascade via SMO and Gli, as well as by mTOR and the JAK/STAT signaling. It promotes cell survival, invasion and migration and impedes apoptosis [163]. A single in vitro study by Chang et al. evaluated the effects of arsenic trioxide on NSCLC and SCLC cell lines and detected a significant downregulation of hedgehog-signaling key-player, Gli1. Furthermore, cancer stem-like cells (i.e., CD133+, SOX2+, Oct4+) were diminished upon treatment with arsenic trioxide [160]. Interestingly, a SOX2-amplified patient demonstrated an outstanding treatment response to SMO-inhibiting sonidegib treatment [82]. The latter aspects leave unsolved questions regarding the possible impact of hedgehog intercalation in SOX2-amplified SCLC patients. 

#### 2.2.8. EZH2 Inhibition in pRB-Deficient and EZH2 Upregulated SCLC

In SCLC, the epigenetic modifier ‘enhancer of zeste homolog 2’ (EZH2) is upregulated upon inactivation of the pRB and E2F-pathway and leads to aberrant methylation of its target, the ‘polycomb repressive complex 2’ (PRC2), ultimately promoting the malignant biology of SCLC [164]. Lately, in n = 40 resected SCLC tissues Toyokawa et al. detected increased expression of EZH2 in two-thirds of the investigated samples [34]. In consideration of the cancer genome atlas, EZH2 gene expression in SCLC is stronger compared to any other neoplasm. Both in vivo and in vitro inhibition of EZH2 decreased tumor growth in SCLC cell lines and patient-derived xenograft mouse models [165]. Potentially, EZH2 inhibition could offer new therapeutic perspectives.

#### 2.2.9. Targeting Hsp90 in SCLC

The ‘heat shock protein 90’ (HSP90) is a 90 kDa sized chaperone that stabilizes its targeting proteins and in SCLC especially, plays a crucial role in longevity, metastasis and resistance to chemotherapy [166]. Here, PEN-866, a HSP90-targeted molecule linked to the irinotecan prodrug SN-38 was evaluated in vitro and in vivo, with promising results [167]. At present, it is under investigation in a phase I/II study (NCT03221400). 

### 2.3. Targeting Errant Signal Cascades in SCLC

#### 2.3.1. mTOR Inhibition and PIK3CA Inhibition in PTEN-Deficient SCLC

Deficiency of ‘phosphatase and tensin homolog’ (PTEN) is a frequent finding in SCLC but not in NSCLC [168]. Sundaresan et al. discovered PTEN deficiencies in 7.4% of SCLC patients [21]. In n = 204 Japanese SCLC tissue samples, alterations in the PIK3CA/AKT/mTOR signaling were analyzed. Mutations for PIK3CA (3%), PTEN (4%) and mTOR1-regulating ‘tuberous sclerosis 1’ (TSC1) (0.5%) were rare. Nevertheless, these patients had poorer survival (Hazard Ratio 2.14), inferior response rates to first-line chemotherapy (30% vs. 53%) and reduced progression-free survival (2.9 months vs. 4.8 months) compared to the control group [169]. The latter data implicate a need to study this therapy in case of alterations of the presented signal cascade further.

PTEN is a tumor-suppressor phosphatase, that dephosphorylates PIK3CA, and therefore, stops downstream signaling to AKT and ultimately, mTOR [170]. In PTEN deficiency, therapeutic targets could be upstream inactivation of PIK3CA or downstream inactivation of mTOR via rapamycin derivatives. Also, activating mutations in PIK3CA could be targeted downstream via mTOR inhibitors.

A single phase I study of the irreversible PIK3CA inhibitor PX-866 in combination with docetaxel in advanced solid malignancies was performed. Here, two patients with SCLC were treated, yet therapeutic implications for SCLC cannot be extracted [171]. The randomized phase II study of PX-866 and docetaxel was performed solely in relapsed or metastatic NSCLC, but failed to improve PFS, ORR or OS [172]. However, treatment of PTEN-deficient SCLC patients was not studied thoroughly. No data for upstream inhibition with PIK3CA-inhibitors such as copanlisib in SCLC exists.

The mTOR inhibitor, everolimus (RAD001), is well-studied in various solid tumors. A phase II study investigated everolimus monotherapy in n = 35 relapsed SCLC patients. Disease control rates of 26% with 3% PR and 23% SD were found, see Table 3. Hence, monotherapy with everolimus is not considered as an appropriate therapy for relapsed SCLC patients [84]. Subsequently, two-phase IB studies evaluated the synergistic use of everolimus for combined therapies. Sun et al. evaluated n = 21 pretreated SCLC patients using a treatment with everolimus (optimal dose 5 mg daily) and paclitaxel, achieving an ORR of 28% [85], see Table 3. Additionally, Besse et al. analyzed the optimum treatment regimen of cisplatin, etoposide, everolimus and ‘granulocyte colony stimulating factor’ (G-CSF) in n = 40 untreated extensive-stage SCLC patients. For an optimal treatment dose of 2.5 mg, everolimus daily plus G-CSF after cisplatin and etoposide, they reported an ORR of 58.3% and a PFS of 35.1 weeks [86], see Table 3. Nevertheless, further clinical evaluations in phase III trials have not been performed.

None of these studies evaluated PTEN deficiency or PIK3CA mutations in the SCLC patient cohort. As depicted, mTOR inhibition by everolimus or temsirolimus, as well as PIK3CA inhibition by copanlisib or PX-866, might prove beneficial in these particular patients.

#### 2.3.2. IGF-1R Inhibition in SCLC

The ‘insulin-like growth factor receptor 1’ (IGF-1R) is frequently overexpressed (71.4%) in SCLC patients. IGF-1R downstream signaling suppresses apoptosis via PI3KCA activation, caspase 3 and BCL-2-family member upregulation (i.e., BCL-2, BAD, BAX and BAK) [173]. Moreover, IGF-1R overexpression in SCLC is associated with dismal outcome, tumor size, higher Ki67 rates and lymphatic spread [174]. Autocrine IGF-1 stimulation in SCLC seems relevant for tumor progression [175]. Against this background, the inhibition of the surface receptor tyrosine kinase IGF-1R offers new treatment options [176]. Unfortunately, a phase II study of platinum-based chemotherapy in combination with either vismodegib (SMO-inhibitor), cixutumumab (IGF-1R antibody) or placebo did not reveal any benefits [83], see Hedgehog-cascade-inhibiting agents in SCLC and Table 3. 

Likewise, a phase II randomized second-line study of linsitinib vs. topotecan in relapsed SCLC failed to show any clinical activity, with a PFS that significantly favored topotecan [87], see Table 3. Figitumumab, another monoclonal antibody associated with IGF-1R, failed to show efficacy in NSCLC [177,178] but clinical data for SCLC is rare [174]. A placebo-controlled study of platinum and etoposide plus figitumumab in SCLC was halted due to low accrual and discontinuation of further clinical investigation of figitumumab after enrolment of n = 9 patients (NCT00977561). 

### 2.4. Targeting Specific Surface Markers in SCLC

As mentioned above, SCLC tissues very likely reveal specific surface markers, such as CD56, DLL3 or TROP-2, see Table 1. Some of these are useful factors for specifically directing cytotoxic agents to SCLC tissues. Figure 3 depicts relevant antibody-drug/cytokine conjugates tested as new therapeutic constructs for SCLC [179,180,181].

#### 2.4.1. CD56-Targeted Therapy in SCLC

CD56 or ‘neural cell adhesion molecule 1’ (NCAM1) is expressed on the surface of neurons and glia cells, as well as natural killer cells, dendritic cells and a subset of T cells (i.e., γ/δ T cells and activated CD8-positive T cells). In the case of malignant disease, CD56 expression can be found in cutaneous Merkel-cell carcinomas, ovarian cancers, myelomas or neuroendocrine tumors, such as SCLC [182]. Its function in SCLC has, to date, not been clarified. In ovarian cancer, the FGFR and NCAM are crucial for cell migration, invasion and metastasis [183]. As CD56 expression is common in SCLC, an antibody-drug conjugate, lorvotuzumab mertansine (IMGN901), was developed and tested. While in murine models, a therapeutic response was observed [182], the addition of lorvotuzumab mertansine to carboplatin and etoposide for the treatment of SCLC with extensive disease stages (phase I/II study) did not improve the outcome, see Table 3. Moreover, in patients receiving lorvotuzumab mertansine, treatment-related severe adverse events (i.e., febrile neutropenia, pneumonia, sepsis) with a lethal outcome occurred in 18 patients [88].

It is possible that CD56 antibody-drug conjugate, promiximab duocarmycin, might show clinical benefits. Up to now, its benefit was shown in patient-derived xenograft mouse models [184]. Yu et al. furthermore, developed an antibody-drug conjugate of promiximab with ‘monomethyl aristatin E’ (MMAE), a microtubule disrupting agent, that showed efficacy in SCLC cell culture and murine models [185].

Apart from antibody-drug conjugates, Crossland et al. evaluated CD56-specific ‘chimeric antigen receptor T cells’ (CD56R CAR T cells) in neuroblastoma and SCLC mouse models. Tumor burden was significantly reduced after infusion of these CAR T cells. However, the survival rate was not improved [186]. Due to expression of CD56 on neurons and glia cells, possible neurotoxic side effects of this treatment should be considered before applying this treatment to humans.

#### 2.4.2. DLL3- and DLL4-Targeted Therapy in SCLC

The ‘delta-like 3’ (DLL3) protein is a surface marker, found in 82.1% of NSCLC patients [17] and in 82.5% of SCLC patients. In SCLC tissues, 31.7% of the specimen revealed high DLL3 expression in ≥50% of the tumor cells [18]. Upon activation, DLL3 serves as an inhibitor of the tumor-suppressor gene, NOTCH1, in SCLC [179,180]. In murine models of Lewis lung carcinoma, high surface levels of DLL3 promoted cell growth and fission, as well as a reduction of apoptosis [17]. Interestingly, mRNA-levels of DLL3 in relapsed or refractory SCLC are higher than in naïve SCLC [187]. Rovalpituzumab tesirine (SC16LD6.5) is an antibody-drug conjugate, which targets the DLL3 surface protein and upon binding and internalization, delivers the cytotoxic agent directly to the DLL3-expressing cell [188]. Saunders et al. analyzed its use in SCLC cell cultures and ‘patient-derived xenograft’ (PDX) mouse models, showing promising anti-tumor effects [187]. Sharma et al. demonstrated a linear association of SC16-uptake and DLL3-expression in SCLC-tissues by ^89^zirconium-tracer labeling [189].

Its use for second- or third-line treatment in both SCLC (n = 74) and LCNEC (n = 8) was evaluated by Rudin et al. In this phase, I–II trial patients with ≥50% DLL3 expression had a better response (ORR 38%) upon Rovalpituzumab tesirine (Rova-T) compared to tumors with <50% DLL3-staining cells (ORR 0%). Likewise, toxicity (i.e., thrombocytopenia, effusions and skin reactions) was predominantly found in patients with a high expression of DLL3 [89], see Table 3. At present, the phase II TRINITY trial assesses third-line treatment of high DLL3 expression in SCLC (NCT02674568). Another phase I study evaluates Rova-T as first-line treatment in combination with cisplatin and etoposide (NCT02819999). Additionally, the TAHOE phase III trial assessed second-line treatment of Rova-T in comparison to standard of care treatment (i.e., topotecan) after progression or relapse upon platinum-based first-line therapy (NCT03061812). However, the TAHOE trial was halted due to inferior outcomes of Rova-T-treated patients.

DLL4 has been analyzed as a target in SCLC in both in vitro and in vivo models. Here, Kuramoto et al. demonstrated that the domain antibody, Dll4-Fc, reduced liver metastasis in mice and led to a down-regulation of NFκB target genes [190]. Another DLL4-directed antibody, MEDI0639, was also tested preclinically and revealed reduced binding of DLL4 to its target structure Notch1, resulting in a reduced neo-angiogenesis [191]. A phase I trial hinted at possible clinical efficacy of MEDI0639 in n = 4 SCLC patients [192] without further clinical investigation.

Two phase I studies evaluated for DLL3-directed T-cell enhancement, either via a bispecific antibody (i.e., AMG757-CD3, NCT03319940) or via CAR T cell therapy (NCT03392064) in refractory and relapsed SCLC patients, who suffered from progressive disease following platinum-based regimens [193]. 

#### 2.4.3. TROP-2-Targeted Therapy in SCLC

The ‘trophoblast cell-surface antigen 2’ (TROP-2) is a surface marker, initially found on trophoblasts, but largely upregulated on epithelial cancer types, such as NSCLC, SCLC, triple-negative breast cancers and urothelial cancers. Its downstream signaling in tumor cells has been reported by calcium-dependent activation of ‘mitogen-activated protein kinase’ (MAPK), by ‘protein-kinase C’ (PKC)-induced activation of RAF and NFκB, as well as interaction with cyclin D1 itself [181]. Gray et al. investigated TROP-2-targeting antibody-drug conjugate, Sacituzumab govitecan (IMMU-132), in n = 50 metastatic and pre-treated SCLC patients (phase II study). Grade 3 and 4 toxicities were neutropenia, fatigue, diarrhea and anemia. In this pretreated subset, treatment outcome with an OS of 7.5 months is encouraging, see Table 3. Interestingly, those patients with prior topoisomerase I-inhibiting treatment with topotecan or irinotecan in particular responded well to the antibody-delivered topoisomerase I inhibitor govitecan (OS 8.8 months) [90]. Besides SCLC, the agent also showed impressive responses in refractory NSCLC patients [194]. However, further investigation of its use in first-line treatment is needed.

#### 2.4.4. EpCAM-Targeted Therapy in SCLC

The role of ‘epithelial cell adhesion molecule’ (EpCAM/ CD326/ TROP-1) in SCLC stem cells was first assumed by Jahcham et al. in Tp53- and pRB-deficient mouse models. They revealed the EpCAM^high^ subset of cells to be highly dependent on the oncogenic MYC driver [195]. Exploring its supplemental driving character in a phase II study of n = 108 SCLC patients at an extensive disease stage was performed. The study conducted a randomization after treatment response to platinum-based chemotherapy, split into best supportive care (BSC) (n = 44) and maintenance therapy (n = 64) with cyclophosphamide and tucotuzumab, an EpCAM-mediated antibody-directed specific IL2 immunocytokine. However, maintenance therapy failed to prove its superiority to BSC. In an exploratory analysis, patients receiving standard-of-care percutaneous cranial irradiation therapy (n = 26) after the end of the primary therapy, beneficial aspects of tucotuzumab and cyclophosphamide were reported [91], see Table 3.

#### 2.4.5. Other Surface Markers in SCLC and SCLC Environments

Lately, preclinical studies on CD133-expressing, etoposide-resistant, tumor stem-like SCLC cells have shown therapeutic potential in the inhibition of neuro-peptide 1 [196]. Moreover, the NGR-peptide, presented in the vascular infarction section of this review, was coupled to ‘human tumor necrosis factor alpha’ (hTNF). Here, NGR is used to link to vessel CD13 and result in local inflammation by hTNF. The concomitant use of doxorubicin and NGR-hTNF in a single-arm phase-II trial of 28 relapsed SCLC patients revealed promising results (PFS 3.2 months) and a manageable safety profile [197].

### 2.5. Targeting Anti-Apoptotic Markers in SCLC

Apoptosis is a crucial event in the course of cancerogenesis. With a focus on the pathophysiology, there are various possibilities yielding the deregulation of apoptosis, see Figure 4. One mechanism occurs due to the increased sensibility of intact cells to caspase activation (similar to tumor suppressors) by nuclear phosphoproteins such as pp32/PHAPI [198]. Another possibility to induce cell death is the activation of the so-called mitochondrial pathway, orchestrated by members of the ‘B-cell leukemia/lymphoma-2’ (BCL-2) protein family [199]. Since BCL-2 is located on the mitochondrial outer membrane, it can control caspase-inducing proteins (e.g., cytochrome c) [200]. Moreover, anti-apoptotic BCL-2 can also inhibit the activation of pro-apoptotic proteins such as BAX or BAK. Thus, BCL-2 family members influence the mitochondrial apoptotic pathway and ultimately, cell death [201,202]. Even though the functional role of BCL-2 seems clear, in NSCLC, its prognostic impact often differs from its anticipated effect. While some studies suggest a positive prognostic effect [203,204,205,206,207], other studies report on negative prognostic effects instead [208,209,210].

#### 2.5.1. BCL-2-Targeted Therapies in SCLC

Proteins of the BCL-2 family are overexpressed in 65–98% of all SCLC patients [22,211,212], especially by refractory cells [213]. Physiologically, BCL-2, BCL-X_L_ and MCL-1 control the outer mitochondrial membrane integrity. In case of disruption, mitochondrial caspases and cytochrome c are released and induce programmed cell death. Apoptotic evasion in various tumors often correlates to BCL-2 overexpression [214]. BCL-2 inhibition with, e.g., venetoclax (ABT-199) reduces the binding of pro-apoptotic ‘BCL-2-like protein 11’ (BCL2L11/BIM) to BCL-2, destabilizing the BCL-2/BIM complex and ultimately, inducing apoptosis [215,216]. Both in vitro and in vivo data suggest therapeutic enhancement of BCL-2 inhibition by interaction with the PI3K/AKT/mTOR pathway via PI3K inhibition [217], mTOR inhibition [218] or via pro-apoptotic Noxa-upregulation by the HDAC inhibitor, vorinostat [219].

A randomized, placebo-controlled phase II study of oblimersen, a BCL-2 antisense oligonucleotide, in n = 56 therapy-naïve SCLC patients with extensive-stage SCLC, in combination with platinum-based chemotherapy, did not show any clinical efficacy with similar ORR (61% vs. 60%) in both study arms. Median OS was even worse, when adding oblimersen to carboplatin and etoposide (8.6 months vs. 10.6 months) [92], see Table 3. Consequently, further clinical evaluation of the antisense nucleotide, oblimersen, was stopped. Unfortunately, the BCL-2 small-molecule inhibitor, navitoclax (ABT-263), also did not reveal any relevant clinical benefit in a single-arm phase IIA study involving relapsed SCLC patients [93], see Table 3. At present, one phase I–II study, with a focus on the mTOR inhibitor, vistusertib, in combination with navitoclax in advanced SCLC is ongoing (NCT03366103). Moreover, two other phase I studies are currently recruiting patients. In both of them, APG-1252, a combined BCL-2/BCL-X_L_ inhibitor, will be investigated in refractory SCLC patients (NCT03387332 and NCT03080311). 

#### 2.5.2. BET Inhibition in MYC-L and MYC-N-Amplified SCLC

‘Bromodomain and extra-terminal domain’ (BET) proteins interfere with acetylated histones and, therefore, work as epigenetic modifiers of gene transcription [220,221]. Kato et al. described in vitro activity of the BET inhibitor, JQ1, in MYC-L-expressing SCLC cell lines. Here, amplification of the MYC-L region might serve as a target of JQ1 [220]. Moreover, Wang et al. found MYC-N-amplified SCLC cell lines respond to the combination of JQ1 with the ‘B-cell lymphoma 2’ (BCL-2) inhibitor navitoclax (ABT-263). Here, BET inhibition via JQ1 induced the expression of ‘BCL-2 like protein 11’ (BCL2L11/ BIM) and thus increased the cellular vulnerability to navitoclax [222]. Lam et al. demonstrated that the BET-inhibitor, mivebresib (ABBV-075), induced the expression of pro-apoptotic BIM, increasing cellular susceptibility to treatment with the BCL-2 inhibitor venetoclax (ABT-199). A xenograft mouse model that proved the cell culture studies and growth suppression via BET- and BCL-2- co-inhibition was shown in vivo [223]. Results of mivebresib in a human phase I study for solid neoplasms with participation of SCLC patients (NCT02391480) are expected [224]. Beyond synergism with BCL-2 inhibitors, BET inhibitors might prove beneficial in combination with HDAC inhibitors, CDK inhibitors and chemotherapy due to their effects on the JAK/STAT pathway, NFκB signaling and p53 acetylation [221,225].

#### 2.5.3. PARP1 Inhibition in SCLC

In 2012, Byers et al. performed proteomic profiling in lung cancer. Amongst EZH2 and ‘B-cell lymphoma 2’ (BCL-2), they determined significantly upregulated mRNA levels of ‘poly-(ADP)-ribose polymerase 1’ (PARP1) in SCLC compared to NSCLC [226]. PARP1 is an enzyme in the nucleus that acts upon DNA-single- or -double-strand-breaks and performs poly-ADP-ribosylation of histones and DNA-repair enzymes. In a malignant setting, it can prevent cells from apoptosis by cleaving DNA-damage and thus inhibits apoptosis induction [227]. Consequently, PARP-inhibition in SCLC appears promising. However, response rates in preclinical studies were disappointing, possibly due to diverging levels of ‘Schlafen-11’ (SLFN11) [228], which is a DNA/RNA helicase, inducing S-phase arrest concomitant to DNA damage by, e.g., cisplatin [229]. Chemotherapeutic resistance in genetically engineered SCLC mouse models was correlated with SLFN11 suppression. Here, EZH2-dependent H3K27me3 histone modification led to SLFN11 gene silencing and, therefore, downregulation of the SLFN11 enzyme. Vice versa, EZH2 inhibition led to elevated SLFN11 levels and recovery of chemosensitivity [230].

A phase I study of veliparib as first-line treatment in combination with carboplatin and etoposide in n = 25 extensive disease stage SCLC patients resulted in an ORR of 64%. The recommended phase II dose of 240 mg veliparib plus carboplatin and etoposide resulted in a response rate of 83% in SCLC, recommending its evaluation for further clinical studies [94], see Table 3. A first-line randomized, placebo-controlled, phase II trial of veliparib plus cisplatin and etoposide in n = 128 extensive-stage SCLC patients prolonged PFS if veliparib was added to the standard of care treatment [95], see Table 3. 

However, despite an improved ORR, the randomized, placebo-controlled phase II study with temozolomid plus veliparib in relapsed or chemorefractory SCLC patients did not show prognostic benefits in the overall cohort. However, in biomarker-assessed high SLFN11 expression, veliparib significantly prolonged PFS and OS [96], as shown in Table 3. The latter aspect proposes clinical studies on EZH2-guided PARP-inhibition and chemotherapy in SCLC. Other than that, a phase I–II study of veliparib and topotecan in refractory or relapsed, platinum-pretreated SCLC is ongoing (NCT03227016).

## 3. Conclusions

Recently, immunotherapy augmented our therapeutic spectrum both for NSCLC [36,37,38,39,40,41] and SCLC patients [43,44]. Besides immune-oncology and chemotherapy, molecular-targeted therapies belong to standard therapy of metastasized NSCLC patients, but not for SCLC patients. To date, no molecular-targeted therapy is approved for SCLC therapy, even though there are distinct pathways and molecular factors such as surface markers, apoptotic proteins, or vascular factors, which might offer new therapeutic options. Among the mentioned therapies, in our opinion there are two promising drugs which might serve for additional clinical evaluation and possibly future therapy: On the one hand, there is the aurora kinase inhibition by alisertib [80,81], on the other hand, there is the PARP-inhibition by veliparib [95]. Besides these, anti-vascular treatment on the basis of tumor vessel infarction could be further studied. Without doubt, poor survival rates of SCLC patients raise the need for new therapeutic principles.

## Figures and Tables

**Figure 1 cancers-11-00690-f001:**
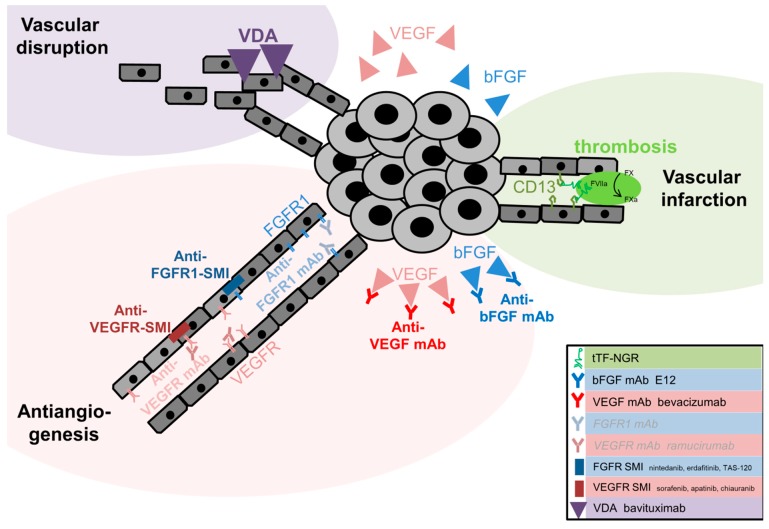
Targeted therapies in the vascular system of SCLC. [97,98,99]. (**a**) Antiangiogenesis: There is an autocrine stimulation of SCLC by the secretion of ‘basic fibroblast growth factor’ (bFGF) and ‘vascular endothelial growth factor’ (VEGF). The latter activates the receptors FGFR and VEGFR and thus induces angiogenesis. Binding of bFGF and VEGF by monoclonal antibodies (mAb) interferes with the signaling cascade. Its receptors may be influenced by monoclonal antibodies or small molecule inhibitors (SMI) such as erdafitinib or TAS-120 for FGFR and apatinib or chiauranib for VEGFR. (**b**) Vascular infarction: The truncated tissue factor-NGR fusion protein binds CD13 on tumor endothelial cells and induces the extrinsic coagulation cascade by activating factor X. (**c**) Vascular disruption: bavituximab binds the phosphatidylserine of tumor vasculature and induces cellular inflammation.

**Figure 2 cancers-11-00690-f002:**
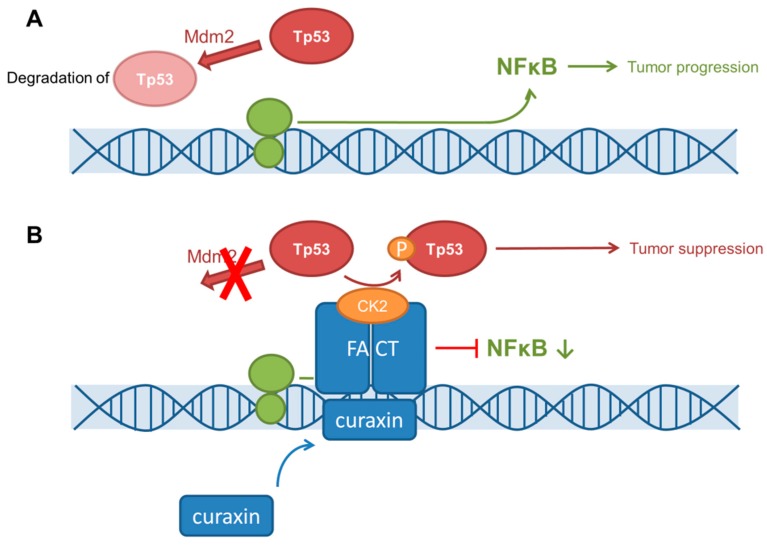
Curaxins’ mechanism of action [145]. (**A**) Without curaxin interaction, Tp53 is degraded by mouse double minute 2 homolog (Mdm2) and the NFκB gene region is transcripted and translated. (**B**) By interaction with curaxin, FACT binds to the DNA double strand and inhibits the transcription of the NFκB gene region. Additionally, casein kinase 2 (CK2) phosphorylates Tp53 and therefore, reduces its degradation.

**Figure 3 cancers-11-00690-f003:**
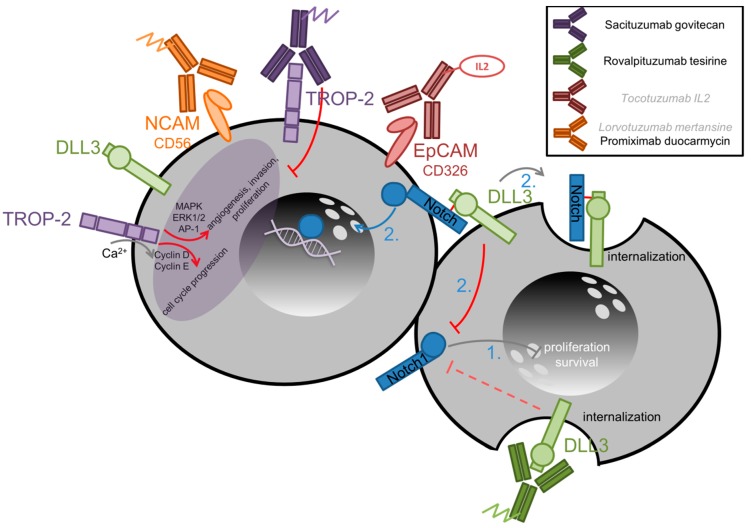
Targeting surface markers and TROP-2 and DLL3 mechanisms of action in SCLC [179,180,181]. Two SCLC cells interact. In SCLC, Notch1 inhibits proliferation and survival (**1.**). Upon DLL3-binding to Notch (**2.**), Notch is cleaved and the ‘Notch intracellular domain’ (NICD) detaches and activates target genes. Moreover, the extracellular Notch compartment is internalized into the DLL3-carrying cell. TROP2 acts via calcium-mediated Cyclin D and E dependent cell cycle control, as well as by activating the ‘mitogen-activated protein kinase’ (MAPK), ‘extracellular signal–regulated kinases’ (ERK), and ‘activator protein 1’ (AP-1) transcription factor signaling cascade.

**Figure 4 cancers-11-00690-f004:**
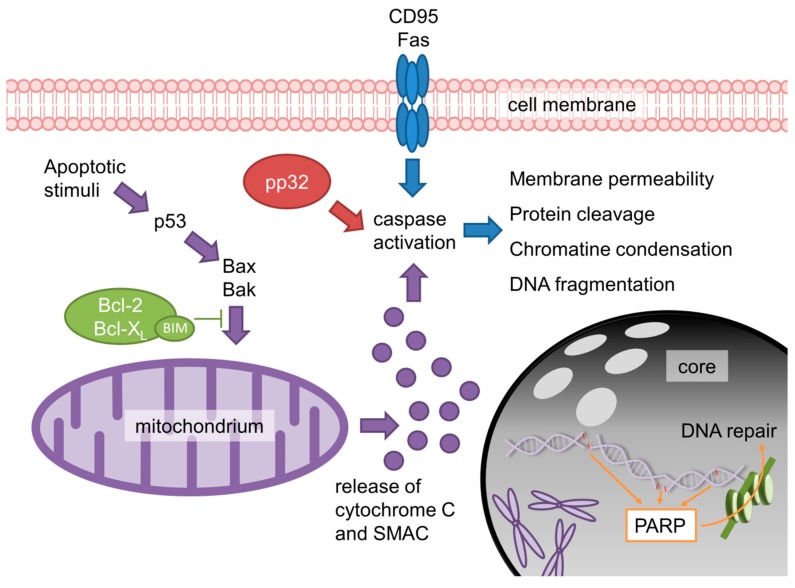
Pro-apoptotic and anti-apoptotic pathways in cancerogenesis. Both extracellular and intracellular pathways lead to caspase activation and thus induce cell death. Distinct factors such as pp32/PHAPI sensitize cells to apoptotic stimuli.

**Table 1 cancers-11-00690-t001:** Frequent mutations and protein expression patterns in small cell lung cancer (SCLC).

Mutation/ Protein expression		Function in SCLC	Frequency	Reference	Possible Targeted Therapies
**TP53**	Inactivation	Tumor suppressor	84.9% (n_tot_ = 272)	[21]	Curaxins
**RB1**	Inactivation	Tumor suppressor	57.4% (n_tot_ = 272)	[21]	AURKi
**NOTCH1**	Inactivation	Tumor suppressor	14.5% (n_tot_ = 110)	[12]	Curaxins
**EP300**	Inactivation	Histone modifier	12.7% (n_tot_ = 110)	[12]	
**CREBBP**	Inactivation	Histone modifier	14.5% (n_tot_ = 110)	[12]	HDACi
**KMT2D/MLL2**	Inactivation	Histone modifier	8.0% (n_tot_ = 40)	[30]	
**PTEN**	Inactivation	AKT inhibitor	7.4% (n_tot_ = 272)	[21]	mTORi/PIK3CAi
**TP73**	Inactivation	Tumor suppressor	12.7% (n_tot_ = 110)	[12]	
**LRP1B**	Inactivation		36.8% (n_tot_ = 272)	[21]	
**MYCL1**	Amplification	Oncogene	6.9% (n_tot_ = 87)	[31]	BETi
**MYCN**	Amplification	Oncogene	4.6% (n_tot_ = 87)	[31]	BETi + BCL2i
**MYC**	Amplification	Oncogene	6.9% (n_tot_ = 87)	[31]	AURKi
**FGFR1**	Amplification	RTK	5.6% (n_tot_ = 251)	[32]	FGFRi
**SOX2**	Amplification	Oncogene	26.8% (n_tot_ = 56)	[33]	Arsenic trioxide
**EZH2**	Expression	Oncogene	62.5% (n_tot_ = 40)	[34]	EZH2i
**CD56/ NCAM1**	Expression	Cell adhesion	95.3% (n_tot_ = 107)	[22]	CAR T cells
**BCL-2**	Expression	Anti-apoptosis	97.6% (n_tot_ = 82)	[22]	BCL-2i
**DLL3**	Expression	NOTCH inactivator	82.5% (n_tot_ = 63)	[18]	Rova-T
**DLL3 high ***			31.7% (n_tot_ = 63)		
*** >50% cellular expression (immunohistochemistry)**					

**Table 2 cancers-11-00690-t002:** Molecular principles for targeted therapies [66,67,68,69,70].

	Monoclonal Antibody (mAb)	Small Molecule Inhibitor (SMI)
**Biology andweight**	Protein two heavy chains (50 kDa each) and two light chains (25 kDa each) with a conserved (F_c_) and an antigen-binding (F_ab_) region	Molecule (<1 kDa)
**Terminology and subclasses**	a) whole antibody*-mab b) antibody–drug conjugate c) antibody–chelator * conjugate (* e.g., carrying radioactive ^90^yttrium) d) domain antibody - fragments of variable singlechains (scFv) - fragments of variable antigen binding sites (Fab)	a) tyrosine kinase inhibitor *-tinib b) RAS-kinase inhibitor *-farnib c) RAF-kinase inhibitor *-rafenib d) angiogenesis inhibitor *-anib e) PI3-kinase inhibitor *-lisib f) proteasome inhibitor *-mib g) cyclin dependent kinase inhibitor *-ciclib h) poly-ADP-ribose polymerase inh. *-parib i) rapamycin derivate; mTOR inh. *-rolimus j) mut. isocitrate dehydrogenase inh. *-denib k) BCL-2 inhibitor *-toclax l) histone deacetylase inhibitor *-inostat m) BET inhibitor *-bresib n) EZH2 inhibitor *-tostat o) hedgehog/SMO inhibitor *-degib
**Targets**	a) cell surface markers, b) kinase receptors, c) binding and deleting antigens.	a) cellular signaling
**Application**	Intravenous (i.v.), subcutaneous (s.c.)	Orally (p.o.), intravenous (i.v.), subcutaneous (s.c.)

**Table cancers-11-00690-t003a:** (**Part I**). Clinical trials evaluating targeted therapies in SCLC.

Reference	Study	Target	Drug	Phase	Setting	Treatment Arms	eval. n	ORR	OS		PFS/TTP/FFS		Result
[71]	Tiseo et al., 2017	VEGF	Bevacizumab	III	ES-SCLC 1st	Cis/Eto + Placebo	103		8.9 m	*p* = 0.113	PFS 5.7 m	*p* = 0.030	
						Cis/Eto + Bevacizumab	101		9.8 m		PFS 6.7 m		-
[72]	Han et al., 2016	VEGF-R, FGFR, PDGF-R	Nintedanib	II	R/R SCLC	Nintedanib mono	22	5%	9.8 m		PFS 1.0 m		-
[73]	Ready et al., 2015	PDGF-R, VEGF-R, RET, c-KIT, FLT3	Sunitinib	II	ES-SCLC 1st	Platin/Eto_4-6_, Placebo maint.Platin/Eto_4-6_, Sunitinib maint.	4649		6.9 m 9.0 m	*p* = 0.160	PFS 2.1 m PFS 3.7 m	*p* = 0.020	+/-
[74]	Sanborn et al., 2017	VEGF-R, EGFR, RET	Vandetanib	II	ES-SCLC 1st	Platin/Eto + Placebo	33	65.4%	9.23 m	*p* = 0.458	TTP 5.62 m	*p* = 0.952	
						Platin/Eto + Vandetanib	33	50.0%	13.24 m		TTP 5.68 m		-
[75]	Abdelraouf et al., 2016	PDGF-R, VEGF-R,	Sunitinib	II	ES-SCLC 1st/	Sunitinib mono	9	11.0%					
		RET, c-KIT, FLT3			relapsed SCLC								-
[76]	Pujol et al., 2007	VEGF-R, FGFR	Thalidomide	III	ES-SCLC 1stadd on	PDCE_2_; Placebo + PDCE_4_	43	84%	8.7 m	*p* = 0.160	PFS 6.4 m	*p* = 0.150	
						PDCE_2_; Thalidomide + PDCE_4_	49	87%	11.7 m		PFS 6.6 m		-
[77]	Lee et al., 2009	VEGF-R, FGFR	Thalidomide	III	LS/ES-SCLC 1st	Carbo/Eto + Placebo	359		10.5 m	*p* = 0.280	PFS 7.6 m	*p* = 0.390	
						Carbo/Eto + Thalidomide	365		10.1 m		PFS 7.6 m		--
[78]	Ellis et al., 2013	VEGF-R, FGFR	Pomalidomide	I	ES-SCLC	Cis/Eto + Pomalidomide	22	31.8%	49.6 w		PFS 12.4 w		-
[79]	Daniel et al., 2017	Notch2, Notch3	Tarextumab	Ib/ II	ES-SCLC 1st	Platin/Eto + Placebo	1371:1		10.3 m	*p* = 0.830	PFS 5.5 m	*p* = 0.940	
						Platin/Eto + Tarextumab			9.3 m		PFS 5.5 m		-
[80]	Melichar et al., 2015	AURK-A	Alisertib	II	R/R SCLC	Alisertib mono	48	21%			PFS 2.1 m		+
[81]	Owonikoko et al., 2017	AURK-A	Alisertib	II	R/R SCLC	Paclitaxel + Placebo	89	18%	165 d	*p* = 0.209	PFS 66 d	*p* = 0.038	
						Paclitaxel + Alisertib	89	22%	186 d		PFS 101 d		+/-
[82]	Pietanza et al., 2016	SMO	Sonidegib	I	ES-SCLC 1st	Cis/Eto + Sonidegib	15	79%	19.7 m		PFS 5.5 m		-
[83]	Belani et al., 2016	SMO IGF-R1	Vismodegib Cixutumumab			Cis/Eto + Placebo	53	48%	8.8 m	*p* = n.s.	PFS 4.4 m		
				II	ES-SCLC 1st	Cis/Eto + Vismodegib	53	56%	9.8 m		PFS 4.4 m	*p* = 0.450	-
						Cis/Eto + Cixutumumab	52	50%	10.1 m		PFS 4.6 m	*p* = 0.480	-
[84]	Tarhini et al., 2010	mTOR	Everolimus	II	R/R SCLC	Everolimus mono	35	3%	6.7 m		TTP 1.3 m		-

**Table cancers-11-00690-t003b:** (**Part II**). Clinical trials evaluating targeted therapies in SCLC.

Reference	Study	Target	Drug	Phase	Setting	Treatment Arms	eval. n	ORR	OS		PFS/TTP/FFS		Result
[85]	Sun et al., 2013	mTOR	Everolimus	Ib	R/R SCLC	Paclitaxel + Everolimus	18	28%					
[86]						Cis/ Eto + d Everol. - GCSF		40.0%					
	Besse et al., 2014	mTOR	Everolimus	Ib	ES-SCLC 1st	Cis/ Eto + w Everol. - GCSF	40	61.1%					−
						Cis/ Eto + d Everol. + GCSF		58.3%			PFS 35.1 w		+/−
[87]	Chiappori et al., 2016	IGF-R1	Linsitinib	II	R/R SCLC	Topotecan mono	15	6.7%	5.3 m	*p* = 0.710	PFS 3.0 m	p < 0.001	
						Linsitinib mono	29	0.0%	3.4 m		PFS 1.2 m		−
[88]	Socinski et al., 2017	CD56	Lorvotuzumabmertansine	I/ II	ES-SCLC 1st	Carbo/ Eto	47	59%	10.1 m	*p* = n.s.	PFS 6.7 m	p = n.s.	
						Carbo/ Eto + Lorvot. mertans.	94	67%	11.0 m		PFS 6.2 m		−
[89]	Rudin et al., 2017	DLL3	Rovalpitu-zumabtesirine	I	R/R SCLC	Rova-T mono overall	60	18%			PFS 2.8 m		
						Rova-T mono - DLL3+	26	38%			PFS 4.3 m		
						Rova-T mono - DLL3-	8	0%			PFS 2.2 m		+
[90]	Gray et al., 2017	TROP-2	Sacituzumabgovitecan	II	R/R ES-SCLC	Sacituzumab govitecan mono	50	14%	7.5 m		PFS 3.7 m		+

[91]	Gladkov et al., 2015	EpCAM	Tucotuzumab	II	ES-SCLC 1stmaintenance	Platin based CTx + BSC	44		14.1 m	*p* = n.s.	PFS 1.4 m	p = n.s.	
						Platin based CTx + Cyc + Tuco.	64		12.3 m		PFS 1.5 m		−
[92]	Rudin et al., 2008	BCL-2	Oblimersen	II	ES-SCLC 1st	Carbo/ Eto + Placebo	15	60%	10.6 m	*p* = 0.020	FFS 7.6 m	p = 0.070	
						Carbo/ Eto + Oblimersen	41	61%	8.6 m		FFS 6.0 m		−
[93]	Rudin et al., 2012	BCL-2	Navitoclax	IIa	R/R SCLC	Navitoclax mono	39	2.6%	3.2 m		PFS 1.5 m		−
[94]	Atrafi et al., 2018	PARP1	Veliparib	I	ES-SCLC 1st	Carbo/Eto + Veliparib overall	25	64%			PFS 5.3 m		
						Carbo/Eto + Veliparib 240 mg	6	83%			PFS 5.6 m		+
[95]	Owonikoko et al., 2018	PARP1	Veliparib	II	ES-SCLC 1st	Cis/Eto + Placebo	64	65.6%	8.9 m	*p* = 0.170	PFS 5.5. m	p = 0.010	
						Cis/Eto + Veliparib	64	71.9%	10.3 m		PFS 6.1 m		+
[96]	Pietanza et al., 2018	PARP1	Veliparib	II	R/R SCLC	Temozolomid + Placebo	52	14%	7.0 m	*p* = 0.500	PFS 2.0 m	p = 0.390	
						Temozolomid + Veliparib	52	39%	8.2 m		PFS 3.8 m		+/−
						Tem. + Placebo - SLFN11+	11		7.5 m	*p* = 0.014	PFS 3.6 m	p = 0.009	
						Tem. + Veliparib - SLFN11+	12		12.2 m		PFS 5.7 m		+

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
