# Peer review of "Future Options of Molecular-Targeted Therapy in Small Cell Lung Cancer"

_cancers, 2019, doi:10.3390/cancers11050690_

Round 1
Reviewer 1 Report
Overall, it is a moderately well written review about the future of molecular targeted therapy in small cell lung cancer. It can be accepted for publication after major revisions to correct mistakes and omissions.
1) Abstract text and main text: I would suggest editing it to state that “standard treatment of SCLC is now platinum and etoposide in combination with atezolizumab” – at least in the US, while chemo doublet alone may be acceptable, it is preferred to combine it with atezolizumab. On March 18, 2019, the Food and Drug Administration approved atezolizumab (TECENTRIQ, Genentech Inc.) in combination with carboplatin and etoposide, for the first-line treatment of adult patients with extensive-stage small cell lung cancer (ES-SCLC), based on overall survival benefit over chemo alone. I would also say in the text that in 2019, FDA has approved this 3-drug combo which is now SoC.
2) Intro: a) please clarify % incidence of SCLC instead of saying “less frequent than NSCLC” b) please add that 3-10% of EGFR-mutant NSCLC treated with EGFR TKIs transform into SCLC. c) when discussing that ROS1 and PIK3CA mutations were detected in 8% of samplles: could break those down separately please? What is the nature of ROS1 mutations? Are these ROS1 translocations? D)Since TKIs are not proven to be benefitial in SCLC, and may never work, could you change your statement from “offers new options for personalized medicine” to “may provide new options..? E) so, talking about IO, a combination of nivolumab and ipilimumab is indeed a treatment option which is one of 3 main options in 2d/3d line SCLC extensive stage therapy in the US, the other two are nivolumab alone, or pembroliuzmab, all listed on US NCCN guidelines web-site. Please cite JNCCN Kalemkerian et al 2018 paper [PMID: 30323087] which enlists 1st line chemo and atezolizumab, as well as ipi-nivo, and pembrolizumab and represents a published version of NCCN guidelines in late 2018. F) It might be worth it to briefly discuss PD-L1 expression and TMB as two potential biomarketrs for SCLC immunotherapy response. Of note, these two, especially the TMB, may be used to predict response to ipi and nivo combination [Hellmann MD et al, Cancer Cell 2018, PMID: 29731394], discussed here in an editorial: PMID: 30233840.
3) Targeted therapies in SCLC text:
a) Please state that targeted therapy works by inhibiting the key driver, usually a specific mutation , where inhibition of a target would result in inhibition of cell growth or cell death.
b) Table 2: column SMI, “Molecule”- please clarify this, is is small / moderate size molecule?
c) Table 1: I would call it more specifically, like “Clinical trials evaluating targeted therapy in advanced SCLC”
d) VEGF and GFG cancerogenic key players: would change that to “key players in cancer progression”
e) F1: I would Capitalize “Targeted therapies…”
f) In section 2.1.1, I would discuss ramucirumab, VEGFR2 mAB / inhibitor which is FDA- approved in NSCLC in 2d line with docetaxel, discussed in PMID: 27200298. Were there trials in SCLC with ramucirumab?
g) Lines 204-205: HJow much was OS improvement for citation #58?
h) 2.2 title: I would rather say “targeting transcription and epigenetic factors in SCLC”
i) Line 262: “solid neoplasms” [plural]
j) Line 319: “did not focus on pts with c-MYC or RB1 status” – please clarify / rephrase what do you mean by status
k) Lane 354-355: I would also mention / cite recent study by Gardner EE et al, Cancer Cell 2017, regarding in vitro EZH2 inhibition.
l) F3: I would add “TROP-2.., DLL3 in SCLC”
m) Lane 434: verum cohort: what’s that?
n) For the Rova-T section, please google search: there were press releases on both, both are halted due to inferior OS with Rova-T [please verify this statement by reading the news] – Rova-T is now dead drug.
o) In the same discussion, you can mention Ambien bi-specific Ab trial [please also define for the readers what is BiTE is], “A Phase 1 Study Evaluating the Safety, Tolerability and Pharmacokinetics of AMG 757 in Subjects With Small Cell Lung Cancer” - AMG 757 is a half-life extended (HLE) BiTE® antibody construct combining the binding specificities for DLL3 and CD3.
p) Suggestion: you could also discuss PEN-866, preclinically tested in PMID: 27267850. This is a conjugate now being investigated in a phase ½ study http://www.tarvedatx.com/PEN-866.html
4) Discussion: would delete DLL3 Rova-T statement since it is now dead drug.
Author Response
Dear editor,
dear reviewer,
thank you very much for reviewing our manuscript entitled "Future options of Molecular targeted Therapy in Small Cell Lung Cancer” (Manuscript ID: cancers-484300). Please find here a point-to-point response.
Comments of Reviewer 1:
1) Abstract text and main text: I would suggest editing it to state that “standard of care treatment of SCLC is now platinum and etoposide in combination with atezolizumab”.
We addressed this issue by adding this information in the abstract section (p. 1, ll. 15-17). “Standard of care treatment of SCLC consists of platinum-based chemotherapy in combination with etoposide lately enhanced by PD-L1 inhibiting atezolizumab in extensive stage disease, as the addition of immune-checkpoint inhibition yielded improved overall survival.”
2) Intro: a) please clarify % incidence of SCLC instead of saying “less frequent than NSCLC”.
We added the SCLC incidence on p. 1, ll. 38-39, now saying “In contrast, with about 14 % small cell lung cancer (SCLC) is less frequent [7] but more aggressive.”
3) Please add that 3-10% of EGFR-mutant NSCLC treated with EGFR TKIs transform into SCLC.
On page 2, ll. 63-64 we provide the information “Of note, about 3 % to 10 % of ‘epidermal growth factor receptor’ (EGFR)-mutant lung adenocarcinoma transform into SCLC [19].”
4) When discussing that ROS1 and PIK3CA mutations were detected in 8% of samples: could you break those down separately please? What is the nature of ROS1 mutations? Are these ROS1 translocations? Since TKIs are not proven to be beneficial in SCLC, and may never work, could you change your statement from “offers new options for personalized medicine” to “may provide new options”?
The COSMIC-browser presents mutations in those gene loci each with 8 %. Nevertheless, were are not able to evaluate the biology of the genetic alterations. The COSMIC browser outputs the information, that in both > 90 % are nonsense or missense substitutions. This hints at untargetable, and maybe clinically inactive products. Therefore we added information on immunohistochemistry of TKIs in SCLC, revealing the seldom incidence of each in SCLC. This section now is presented as follows (p. 2, ll. 81 ff.):
“Following the ‘Catalogue Of Somatic Mutations In Cancer’ (COSMIC) browser in SCLC, other than these, ROS1 mutations and PIK3CA mutations each were detected in 8 % of the samples. However, over 90 % of these were either nonsense or missense substitutions [24]. Regarding immunohistochemistry, receptor tyrosine kinase overexpression, other than c-KIT in LCNEC and SCLC was scarce [25]. Hence, inhibition of tyrosine kinases in SCLC might provide options for personalized medicine in the context of Next Generation Sequencing (NGS) results for instance [26]–[29]”.
5) So, talking about IO, a combination of nivolumab and ipilimumab is indeed a treatment option which is one of 3 main options in 2nd/3rd line SCLC extensive stage therapy in the US, the other two are nivolumab alone, or pembroliuzmab, all listed on US NCCN guidelines web-site. Please cite JNCCN Kalemkerian et al 2018 paper [PMID: 30323087] which enlists 1st line chemo and atezolizumab, as well as ipi-nivo, and pembrolizumab and represents a published version of NCCN guidelines in late 2018.
To address this issue, we added this valuable information the introduction section regarding immunoncology (p. 3, ll. 104-107): “Following the NCCN guidelines [43], based on the study performed by Hellmann et al. the PD-1 inhibitor nivolumab plus the ‘cytotoxic T-lymphocyte associated protein 4’ (CTLA-4) inhibitor ipilimumab is an approved therapy for refractory or relapsed SCLC [44].” However, we did not mention pembrolizumab, as the NCCN guideline by Kalemkerian provides the following information: “Note that pembrolizumab is not currently recommended in the NCCN Guidelines”.
6) It might be worth it to briefly discuss PD-L1 expression and TMB as two potential biomarkers for SCLC immunotherapy response. Of note, these two, especially the TMB, may be used to predict response to ipi and nivo combination [Hellmann MD et al, Cancer Cell 2018, PMID: 29731394], discussed here in an editorial: PMID: 30233840.
We used the proposal of reviewer no. 2 to integrate a section on sex differences in treatment and outcome of lung cancer patients to include this discussion and introduce the immune-checkpoint inhibition on p. 2 l. 88-p. 3 l.97 as follows: “The rising incidence of female lung cancer patients [1], [5] brings forth the question if sex steroids influence tumor progression and ultimately patient’s outcome. Immunohistochemistry in NSCLC patients revealed both high estrogen receptor levels and aromatase enzyme levels in female patients. Here, aromatase enzyme upregulation was associated with reduced outcome [30]. However, there is no evidence for SCLC at present. Interestingly, when it comes to treatment success of immune-checkpoint inhibition the positive tumoral ‘programmed cell death ligand 1’ (PD-L1) stain as well as the ‘tumor mutational burden’ (TMB) is discussed [31]. Here, Wang et al. were able to show, that female NSCLC patients revealed a significantly higher TMB than male patients, predicting better response rates to immune-checkpoint inhibition [32]. Yet, underlying data represent results from NSCLC analysis exclusively and prospectively have to be evaluated in SCLC.”
7) Please state that targeted therapy works by inhibiting the key driver, usually a specific mutation , where inhibition of a target would result in inhibition of cell growth or cell death.
To address this issue, we added the following sentence in section 2, “Targeted therapies in SCLC treatment” on p. 5, ll. 139-141: “Hereby, targeted therapies either interact with key tumor drivers (i.e. mostly specific mutations) that promote cell growth as well as immortality or address tumor specific intra- or extracellular features.”
8) Table 2: column SMI, “Molecule”- please clarify this, is it small / moderate size molecule?
In due consideration of the work of Arkin et al., 2014 we added the information, that the presented small molecule inhibitors are < 1 kDa.
9) Table 1: I would call it more specifically, like “Clinical trials evaluating targeted therapy in advanced SCLC”.
We switched the title of Table 1 to the proposed title.
10) VEGF and FGF cancerogenic key players: would change that to “key players in cancer progression”
We changed the exact wording on page 8, l. 156 to “Among these factors, ‘vascular endothelial growth factor’ (VEGF) and ‘basic fibroblast growth factor’ (bFGF) are key players in cancer progression.”
11) F1: I would Capitalize “Targeted therapies…”
We capitalized “Targeted therapies” in the figure caption.
12) In section 2.1.1, I would discuss ramucirumab, VEGFR2 mAB / inhibitor which is FDA- approved in NSCLC in 2d line with docetaxel, discussed in PMID: 27200298. Were there trials in SCLC with ramucirumab?
To address this issue, we added a small section before addressing the VEGF-R2 SMI apatinib on p. 9, ll. 190-192: “Even though, VEGF-R2 inhibition with Ramucirumab is an accepted therapy for relapsed NSCLC [104], there is little evidence in SCLC. At present, there are ongoing studies, which evaluate both the VEGF-R2 TKI apatinib (NCT02995187) and […]”
13) Lines 204-205: How much was OS improvement for citation #71 (Pujol et al., 2007)?
To address this issue, next to Table 3, now the text presents the data on overall survival on p. 10, ll. 228-233 as follows: “Nevertheless, the placebo-controlled phase III study discovered no significant survival benefit upon the additional application of thalidomide following response to two cycles platinum based chemotherapy (Table 3). Here, the insignificant but impressive prolongation of the overall survival of three months in the thalidomide cohort (i.e. 11.7 months vs. 8.7 months, p=0.160) has to be considered carefully in the light of the study design with its necessary therapeutic response after two cycles of a non-standard of care therapy [71].”
14) 2.2 title: I would rather say “targeting transcription and epigenetic factors in SCLC”
We changed the title to the more precise term “2.2.Targeting transcription and epigenetic factors in SCLC” (p. 10, l. 267)
15) Line 262: “solid neoplasms” [plural]
We changed this spelling mistake on p. 11, l. 289 into “Currently, a phase I trial of CBL0137 in advanced solid neoplasms and glioblastoma (NCT01905228) is ongoing.”
16) Line 319: “did not focus on pts with c-MYC or RB1 status” – please clarify / rephrase what do you mean by status.
To address this issue, we corrected the wording on p. 12, ll. 345-346 into “[…]and both studies did not focus on patients’ c-MYC or RB1 mutational status.”
17) Lane 354-355: I would also mention / cite recent study by Gardner EE et al, Cancer Cell 2017, regarding in vitro EZH2 inhibition.
We addressed this issue in the PARP-section on p. 19, ll. 601 ff. as SLFN-11 recovery by EZH2 inhibition is mentioned there.
18) F3: I would add “TROP-2.., DLL3 in SCLC”
We corrected the figure heading on p. 15, l. 453 to “Targeting surface markers and TROP-2 and DLL3 mechanisms of action in SCLC”.
19) Lane 434: verum cohort: what’s that?
To be more precisely here, we changed the wording on p. 16 ll. 470-472 into “Moreover, in patients receiving lorvotuzumab mertansine, treatment related severe adverse events (i.e. febrile neutropenia, pneumonia, sepsis) with lethal outcome occurred in 18 patients [83].”
20) For the Rova-T section, please google search: there were press releases on both, both are halted due to inferior OS with Rova-T [please verify this statement by reading the news] – Rova-T is now a dead drug.
Thanks in advance for this highly relevant input on Rova-T therapy. We addressed this issue by changing the section on the phase III TAHOE trial on p. 16, ll. 501-504, now reading: “Additionally, the TAHOE phase III trial assessed second-line treatment of Rova-T in comparison to standard of care (i.e. topotecan) after progression or relapse upon platinum-based first line therapy (NCT03061812). However, the TAHOE trial was halted due to inferior outcomes of Rova-T treated patients.”
21) In the same discussion, you can mention Ambien bi-specific Ab trial [please also define for the readers what is BiTE is], “A Phase 1 Study Evaluating the Safety, Tolerability and Pharmacokinetics of AMG 757 in Subjects With Small Cell Lung Cancer” - AMG 757 is a half-life extended (HLE) BiTE® antibody construct combining the binding specificities for DLL3 and CD3.
We added information on the bispecific antibody to the CAR T cell section on p. 16, ll. 511-513 as follows: “Two phase I studies evaluate for DLL3-directed T-cell enhancement, either via a bispecific antibody (i.e. AMG757-CD3, NCT03319940) or via CAR T cell therapy (NCT03392064) in refractory and relapsed SCLC patients, who suffered from progressive disease following platinum-based regimens [189].”
22) Suggestion: you could also discuss PEN-866, preclinically tested in PMID: 27267850. This is a conjugate now being investigated in a phase ½ study http://www.tarvedatx.com/PEN-866.html
We addressed this issue by adding a subsection on HSP90 directed treatments in the topic “2.2. Targeting transcription and epigenetic factors in SCLC” on p. 13, ll. 387-394: “2.2.9.Targeting Hsp90 in SCLC - The ‘heat shock protein 90’ (HSP90) is a 90 kDa sized chaperone that stabilizes its targeting proteins and especially in SCLC plays a crucial role in longevity, metastasis and resistance to chemotherapy [161]. Here, PEN-866, a HSP90-targeted molecule linked to the irinotecan prodrug SN-38 was evaluated in vitro and in vivo with promising results [162]. At present, it is under investigation in a phase I/II study (NCT03221400). Other than that, Ganetespib, a HSP90 inhibitor was evaluated combined with doxorubicin in a phase Ib/II study [163] and also might offer further therapeutic options.”
23) Discussion: would delete DLL3 Rova-T statement since it is now a dead drug.
Consequently, we deleted the section on Rova-T in our conclusion, formerly reading “DLL3 directed Rovalpituzumab tesirine [84],”. On page 20 ll. 633 – 636 it is depicted as follows: “Among the mentioned therapies, in our opinion there are two promising drugs, which might serve for additional clinical evaluation and possibly future therapy: on the one hand there is the aurora kinase inhibition by alisertib [75], [76], on the other hand there is the PARP-inhibition by veliparib [90].”
In case you have further comments or questions, please do not hesitate to contact us.
With our best regards,
PD Dr. Lars Henning Schmidt Dr. Arik Schulze
Reviewer 2 Report
The manuscript by Dr. Schulze and Colleagues is focused on small cell lung cancer (SCLC), which is still considered the more aggressive one.
The authors revise different features of this malignancy and describe in depth the most frequent mutations, protein expression patterns and targeted therapies in SLCS treatment.
This review is fluent and well written, the English language is very good and the graphical art is very well performed. The Tables presented are appropriate and they help the reader.
I have just few curiosities, below attached, and I would like to see in the revised manuscript some comments about these issue:
1) Considering that lung cancer incidence is significantly increased during the last year in women, what is the state of art about the role of sex steroids and sex steroid receptors in lung cancer? Do the steroid receptors have a possible prognostic, diagnostic and therapeutic relevance, in association with survival and therapeutic response of patients?
2) As in part described in this manuscript, targeted therapy and immunotherapy became pillars in lung cancer treatment. What is the impact of radiotherapy, if so, in SCLC? And what about combinatorial radiotherapy plus chemotherapy or immunotherapy ?
3) Use of statins strongly decreases the cholesterol and LDL content. As such, they can affect the risk and prognosis of SCLC?
These questions are related to very recent topics. They could be a starting point for a wider analysis of lung cancer that could attract more audience.
Author Response
Dear editor,
dear reviewer,
thank you very much for reviewing our manuscript entitled "Future options of Molecular targeted Therapy in Small Cell Lung Cancer” (Manuscript ID: cancers-484300). Please find here a point-to-point response.
Comments of Reviewer 2:
1) Considering that lung cancer incidence is significantly increased during the last year in women, what is the state of art about the role of sex steroids and sex steroid receptors in lung cancer? Do the steroid receptors have a possible prognostic, diagnostic and therapeutic relevance, in association with survival and therapeutic response of patients?
To introduce this topic, we added information on rising female incidence rates in lung cancer on p. 1, ll. 35-36: “Interestingly, the female proportion of lung cancer patients has increased especially in Northern America [5], Northern Europe and Australia [1].” thereby citing the paper of Jemal et al., 2018 NEJM.
Later on we then introduced the interesting topic of sex steroids and sex dependent outcome by addressing an issue of reviewer no. 1 – the tumor mutational burden. On p. 2-3, ll. 88-97 there is now the section: “The rising incidence of female lung cancer patients [1], [5] brings forth the question if sex steroids influence tumor progression and ultimately patient’s outcome. Immunohistochemistry in NSCLC patients revealed both high estrogen receptor levels and aromatase enzyme levels in female patients. Here, aromatase enzyme upregulation was associated with reduced outcome [30]. However, there is no evidence for SCLC at present. Interestingly, when it comes to treatment success of immune-checkpoint inhibition the positive tumoral ‘programmed cell death ligand 1’ (PD-L1) stain as well as the ‘tumor mutational burden’ (TMB) is discussed [31]. Here, Wang et al. were able to show, that female NSCLC patients revealed a significantly higher TMB than male patients, predicting better response rates to immune-checkpoint inhibition [32]. Yet, underlying data represent results from NSCLC analysis exclusively and prospectively have to be evaluated in SCLC.”
2) As in part described in this manuscript, targeted therapy and immunotherapy became pillars in lung cancer treatment. What is the impact of radiotherapy, if so, in SCLC? And what about combinatorial radiotherapy plus chemotherapy or immunotherapy?
To address this issue, in brief we added the latest updates on radiotherapy and ‘abscopal effect’ of combination therapies on p. 3, ll. 111-119: “It is known, that ‘prophylactic cranial irradiation’ (PCI) in SCLC is recommended in limited as well as in extensive stage SCLC disease. Regarding its toxicity, experts though have lately discussed its use in stage IV disease without ‘central nervous system’ (CNS) metastasis. In this specific situation, expert opinion limited the use to young and fit patients, that well responded to first line therapy [47]. Other than that, thoracic irradiation in stage I-III disease is improving the prognosis, if started within 30 days to chemotherapy [48]. With regard to the ‘abscopal effect’, local irradiation and systemic chemotherapy in stage IV disease might promote the effect of concomitant immunotherapy with PD-1/ PD-L1 inhibitors by inducing immunogenic cell death [49]. Hence, combination therapies of chemotherapy, irradiation and immunotherapy will be focus of clinical research to further improve SCLC patient’s outcome.”
3) Use of statins strongly decreases the cholesterol and LDL content. As such, can they affect the risk and prognosis of SCLC?
We widened this topic to give information on lipid metabolism and the use of anticoagulants in SCLC patients on page 3, ll. 121-125: “Next to treatment of the tumor entity by operation, irradiation, chemotherapy or immunotherapy, systemic risk factors influence the outcome of SCLC patients. As venous thrombosis and thromboembolism are risk factors for dismal outcome, one study group focused on the additional use of low-molecular weight heparin anticoagulants in SCLC treatment. Yet, neither was overall survival improved by the concomitant use of enoxaparin [50] nor was a biomarker established for predicting the risk of venous thrombosis and thromboembolism in SCLC patients [51]. Other than that, redox status and lipid metabolism was shown to be altered in lung cancer patients. Here, observational studies of lipid metabolism in lung cancer patients revealed reduced levels of high density lipoprotein and apoprotein A1 as well as elevated levels of triglycerides [52]. Nevertheless, the anti-inflammatory and cholesterol-lowering effect of pravastatin plus cisplatin and etoposide did not result in superior outcome in SCLC patients [53].”
In case you have further comments or questions, please do not hesitate to contact us.
With our best regards,
PD Dr. Lars Henning Schmidt Dr. Arik Schulze
Round 2
Reviewer 1 Report
Thank you for improving the manuscript by addressing some of the issues. However, some key problems were not corrected and additional incorrect statements were put in, requiring second major revision. Please correct those more carefully, otherwise I will have to simply reject your manuscript on the next round of revision.
1) When discussing that ROS1 and PIK3CA mutations were detected in 8% of samples: could you break those down separately please? What is the nature of ROS1 mutations? Are these ROS1 translocations? Since TKIs are not proven to be beneficial in SCLC, and may never work, could you change your statement from “offers new options for personalized medicine” to “may provide new options”?
- could you please delete ROS1 mutations statements please, leaving only PIK3CA? I went into COSMIC, and also reviewed the literature. All mutations in ROS1 are most likely just secondary for NSCLC patients with ROS1 translocations treated with crizotinib, so SECONDARY to crizotinib. Further, ROS1 fusions are known to be oncogenic in NSCLC, but I am not aware of any mutations in ROS1 data that can make you thing that those mutations are driver mutations in lung cancer. So, to mention those ROS1 mutations in you review is misleading to the reader, and makes no sense to me at all.
2) So, talking about IO, a combination of nivolumab and ipilimumab is indeed a treatment option which is one of 3 main options in 2nd/3rd line SCLC extensive stage therapy in the US, the other two are nivolumab alone, or pembroliuzmab, all listed on US NCCN guidelines web-site. Please cite JNCCN Kalemkerian et al 2018 paper [PMID: 30323087] which enlists 1st line chemo and atezolizumab, as well as ipi-nivo, and pembrolizumab and represents a published version of NCCN guidelines in late 2018.
A) “Following the NCCN guidelines [43], based on the study performed by Hellmann et al. the PD-1 inhibitor nivolumab plus the ‘cytotoxic T-lymphocyte associated protein 4’ (CTLA-4) inhibitor ipilimumab is an approved therapy for refractory or relapsed SCLC [44].”
- this statement is incorrect, since NCCN has put in nivo +/- ipi recommendation much earlier, based on Antonia SJ et al Lancet Oncology 2016 study, PMID: 27269741, and NOT Hellman et al later publication. Please correct it accordingly.
B) "However, we did not mention pembrolizumab, as the NCCN guideline by Kalemkerian provides the following information: “Note that pembrolizumab is not currently recommended in the NCCN Guidelines”.
-OK, so please do mention pembrolizumab since it is indeed now recommended by the v NCCN since around December 2018. So JNCCN article was published > 6 months ago, and shortly thereafter NCCN has added pembro to the guidelines. Current online version of NCCN, https://www.nccn.org/professionals/physician_gls/pdf/sclc.pdf, version 01.2019 has pembrolizumab listed now based on Chung HC et al ASCO abstract, a cohort of pembrolizumab KEYNOTE-158 study, https://ascopubs.org/doi/abs/10.1200/JCO.2018.36.15_suppl.8506
3) It might be worth it to briefly discuss PD-L1 expression and TMB as two potential biomarkers for SCLC immunotherapy response. Of note, these two, especially the TMB, may be used to predict response to ipi and nivo combination [Hellmann MD et al, Cancer Cell 2018, PMID: 29731394], discussed here in an editorial: PMID: 30233840.
- "Interestingly, when it comes to treatment success of immune-93 checkpoint inhibition the positive tumoral ‘programmed cell death ligand 1’ (PD-L1) stain as well as 94 the ‘tumor mutational burden’ (TMB) is discussed [31]."
- so, the way you added this citation, and discussed it, is misleading. Also it is apparent that you have not reviewed or not read this paper well enough. - Please add that patients with high TMB have demonstrated durable responses to nivolumab and ipilimumab combination, therefore TMB could be a useful biomarker of immunotherapy response in SCLC [Hellman MD et al, Cancer Cell 2018].
4) Suggestion: you could also discuss PEN-866, preclinically tested in PMID: 27267850. This is a conjugate now being investigated in a phase ½ study http://www.tarvedatx.com/PEN-866.html
-Thank you for discussing PEN-866. However, ganetespib is a dead drug, since it failed in Phase 3, and that failure has lead to a collapse of Synta pharmaceuticals, therefore this drug is no longer investigated in lung cancer. - So, please remove "Other than that, Ganetespib, a HSP90 inhibitor was evaluated combined with doxorubicin in a phase Ib/II study [163] and also might offer further therapeutic options.”
Author Response
Dear editor,
Dear reviewer,
thank you very much for reviewing our manuscript entitled "Future options of Molecular targeted Therapy in Small Cell Lung Cancer” (Manuscript ID: cancers-484300). In due consideration of the reviewer’s comments we performed a point-to-point response.
Comments of Reviewer 1:
Thank you for improving the manuscript by addressing some of the issues. However, some key problems were not corrected and additional incorrect statements were put in, requiring second major revision. Please correct those more carefully, otherwise I will have to simply reject your manuscript on the next round of revision.
1) When discussing that ROS1 and PIK3CA mutations were detected in 8% of samples: could you break those down separately please? What is the nature of ROS1 mutations? Are these ROS1 translocations? Since TKIs are not proven to be beneficial in SCLC, and may never work, could you change your statement from “offers new options for personalized medicine” to “may provide new options”?
- could you please delete ROS1 mutations statements please, leaving only PIK3CA? I went into COSMIC, and also reviewed the literature. All mutations in ROS1 are most likely just secondary for NSCLC patients with ROS1 translocations treated with crizotinib, so SECONDARY to crizotinib. Further, ROS1 fusions are known to be oncogenic in NSCLC, but I am not aware of any mutations in ROS1 data that can make you thing that those mutations are driver mutations in lung cancer. So, to mention those ROS1 mutations in you review is misleading to the reader, and makes no sense to me at all.
To address this issue, we removed the ROS1 mutations in the introduction section (p. 2, ll. 82-83), now reading:
“Following the ‘Catalogue Of Somatic Mutations In Cancer’ (COSMIC) browser in SCLC, other than these, ROS1 mutations and PIK3CA mutations each were detected in 8 % of the samples. […]”
2) So, talking about IO, a combination of nivolumab and ipilimumab is indeed a treatment option which is one of 3 main options in 2nd/3rd line SCLC extensive stage therapy in the US, the other two are nivolumab alone, or pembroliuzmab, all listed on US NCCN guidelines web-site. Please cite JNCCN Kalemkerian et al 2018 paper [PMID: 30323087] which enlists 1st line chemo and atezolizumab, as well as ipi-nivo, and pembrolizumab and represents a published version of NCCN guidelines in late 2018.
A) “Following the NCCN guidelines [43], based on the study performed by Hellmann et al. the PD-1 inhibitor nivolumab plus the ‘cytotoxic T-lymphocyte associated protein 4’ (CTLA-4) inhibitor ipilimumab is an approved therapy for refractory or relapsed SCLC [44].”
- this statement is incorrect, since NCCN has put in nivo +/- ipi recommendation much earlier, based on Antonia SJ et al Lancet Oncology 2016 study, PMID: 27269741, and NOT Hellman et al later publication. Please correct it accordingly.
B) "However, we did not mention pembrolizumab, as the NCCN guideline by Kalemkerian provides the following information: “Note that pembrolizumab is not currently recommended in the NCCN Guidelines”.
-OK, so please do mention pembrolizumab since it is indeed now recommended by the v NCCN since around December 2018. So JNCCN article was published > 6 months ago, and shortly thereafter NCCN has added pembro to the guidelines. Current online version of NCCN, https://www.nccn.org/professionals/physician_gls/pdf/sclc.pdf, version 01.2019 has pembrolizumab listed now based on Chung HC et al ASCO abstract, a cohort of pembrolizumab KEYNOTE-158 study, https://ascopubs.org/doi/abs/10.1200/JCO.2018.36.15_suppl.8506
3) It might be worth it to briefly discuss PD-L1 expression and TMB as two potential biomarkers for SCLC immunotherapy response. Of note, these two, especially the TMB, may be used to predict response to ipi and nivo combination [Hellmann MD et al, Cancer Cell 2018, PMID: 29731394], discussed here in an editorial: PMID: 30233840.
- "Interestingly, when it comes to treatment success of immune-93 checkpoint inhibition the positive tumoral ‘programmed cell death ligand 1’ (PD-L1) stain as well as 94 the ‘tumor mutational burden’ (TMB) is discussed [31]."
- so, the way you added this citation, and discussed it, is misleading. Also it is apparent that you have not reviewed or not read this paper well enough. - Please add that patients with high TMB have demonstrated durable responses to nivolumab and ipilimumab combination, therefore TMB could be a useful biomarker of immunotherapy response in SCLC [Hellman MD et al, Cancer Cell 2018].
To address the reviewer’s comment, we modified the immune-oncology part in the introduction section. The paragraph on p. 3, ll. 95-114 was changed as follows:
“Inhibition of ‘programmed cell death 1’ (PD-1)/ PD-L1 axis by nivolumab, pembrolizumab (both PD-1 inhibitors) or atezolizumab (PD-L1 inhibitor) augmented first- and second-line therapeutic options for NSCLC patients [36]–[42]. Recently, the PD-L1 inhibitor atezolizumab proved efficacy for the first-line treatment of extensive-stage SCLC in combination with carboplatin and etoposide [43]. Similar, the PD-1 inhibitor nivolumab yielded therapeutic response in third line SCLC therapy [44]. However, compared to second line topotecan or amrubicin, nivolumab monotherapy application did not result in superior outcomes in SCLC [45]. Here, nivolumab plus the ‘cytotoxic T-lymphocyte associated protein 4’ (CTLA-4) inhibitor ipilimumab is an advised therapy for refractory or relapsed SCLC [46]. Similar, pembrolizumab proved efficacy in refractory or relapsed SCLC [47] and is therefore recommended for SCLC treatment by the National Comprehensive Cancer Network (NCCN) [48].
In NSCLC, positive PD-L1 expression is associated with tumor response to immune-checkpoint-inhibition [36], [49]. Likewise, a positive PD-L1 expression (≥ 1 %) in SCLC is discussed to predict therapeutic response following immune oncologic treatment [46], [47], [50], [51]. Additionally in both SCLC and NSCLC, a high ‘tumor mutational burden’ (TMB) indicates durable treatment responses, especially following nivolumab and ipilimumab combination treatment [51]–[53]. Here, with regard to gender specifics, Wang et al. demonstrated, that female NSCLC patients have a higher predictive value by TMB analysis in comparison to male patients [54]. However, these observations still need to be confirmed for SCLC.”
4) Suggestion: you could also discuss PEN-866, preclinically tested in PMID: 27267850. This is a conjugate now being investigated in a phase ½ study http://www.tarvedatx.com/PEN-866.html
-Thank you for discussing PEN-866. However, ganetespib is a dead drug, since it failed in Phase 3, and that failure has lead to a collapse of Synta pharmaceuticals, therefore this drug is no longer investigated in lung cancer. - So, please remove "Other than that, Ganetespib, a HSP90 inhibitor was evaluated combined with doxorubicin in a phase Ib/II study [163] and also might offer further therapeutic options.”
To address this issue, we removed the proposed sentence on page 13 (ll. 388-394):
“The ‘heat shock protein 90’ (HSP90) is a 90 kDa sized chaperone that stabilizes its targeting proteins and especially in SCLC plays a crucial role in longevity, metastasis and resistance to chemotherapy [166]. Here, PEN-866, a HSP90-targeted molecule linked to the irinotecan prodrug SN-38 was evaluated in vitro and in vivo with promising results [167]. At present, it is under investigation in a phase I/II study (NCT03221400). Other than that, Ganetespib, a HSP90 inhibitor was evaluated combined with doxorubicin in a phase Ib/II study [168] and also might offer further therapeutic options.”
In case you have further comments or questions, please do not hesitate to contact us.
With our best regards,
PD Dr. Lars Henning Schmidt Dr. Arik Schulze

Round 3
Reviewer 1 Report
I accept this latest version. Minor English spelling checks and editing is recommended.
Author Response
Dear Editor,
as recommended our manuscript entitled "Future options of Molecular targeted Therapy in Small Cell Lung Cancer" was submitted to MDPI for English editing.
Plerase, find here the current version after English editing by MDPI.
In case of questions, please do not hesitate to contact us.
With our best regards Arik Schulze and Lars Henning Schmidt